# Efficient Meta Neural Heuristic for Multi-Objective Combinatorial Optimization

**Jinbiao Chen**[1], **Jiahai Wang**[1,2,3,*], **Zizhen Zhang**[1,*], **Zhiguang Cao**[4], **Te Ye**[1], **Siyuan Chen**[1]

[1]School of Computer Science and Engineering, Sun Yat-sen University, P.R. China
[2]Key Laboratory of Machine Intelligence and Advanced Computing, Ministry of Education, Sun Yat-sen University, P.R. China
[3]Guangdong Key Laboratory of Big Data Analysis and Processing, Guangzhou, P.R. China
[4]School of Computing and Information Systems, Singapore Management University, Singapore
chenjb69@mail2.sysu.edu.cn, {wangjiah,zhangzzh7}@mail.sysu.edu.cn
zgcao@smu.edu.sg, {yete,chensy47}@mail2.sysu.edu.cn

## Abstract

Recently, neural heuristics based on deep reinforcement learning have exhibited promise in solving multi-objective combinatorial optimization problems (MO-COPs). However, they are still struggling to achieve high learning efficiency and solution quality. To tackle this issue, we propose an efficient meta neural heuristic (EMNH), in which a meta-model is first trained and then fine-tuned with a few steps to solve corresponding single-objective subproblems. Specifically, for the training process, a (partial) architecture-shared multi-task model is leveraged to achieve parallel learning for the meta-model, so as to speed up the training; meanwhile, a scaled symmetric sampling method with respect to the weight vectors is designed to stabilize the training. For the fine-tuning process, an efficient hierarchical method is proposed to systematically tackle all the subproblems. Experimental results on the multi-objective traveling salesman problem (MOTSP), multi-objective capacitated vehicle routing problem (MOCVRP), and multi-objective knapsack problem (MOKP) show that, EMNH is able to outperform the state-of-the-art neural heuristics in terms of solution quality and learning efficiency, and yield competitive solutions to the strong traditional heuristics while consuming much shorter time.

## 1 Introduction

Multi-objective combinatorial optimization problems (MOCOPs) [1] are widely studied and applied in many real-world sectors, such as telecommunication, logistics, manufacturing, and inventory. Typically, an MOCOP requires the simultaneous optimization of multiple conflicting objectives, where the amelioration of an objective may lead to the deterioration of others. Therefore, a set of trade-off solutions, known as *Pareto-optimal* solutions, are usually sought for MOCOPs.

Generally, it is difficult to exactly find all the Pareto-optimal solutions of an MOCOP [2], especially given that the decomposed single-objective subproblem might already be NP-hard. Hence, heuristic methods [3] are usually preferred to solve MOCOPs in reality, as they can attain approximate Pareto-optimal solutions in (relatively) reasonable time. Nevertheless, the traditional heuristics are still lacking due to their reliance on handcrafted rules and massive iterative steps, and superfluous computation even for instances of the same (or similar) class.

Recently, inspired by the success of deep reinforcement learning (DRL) in learning neural heuristics for solving the single-objective combinatorial optimization problems (COPs) [4–7], a number of

---

*Jiahai Wang and Zizhen Zhang are the corresponding authors.

37th Conference on Neural Information Processing Systems (NeurIPS 2023).

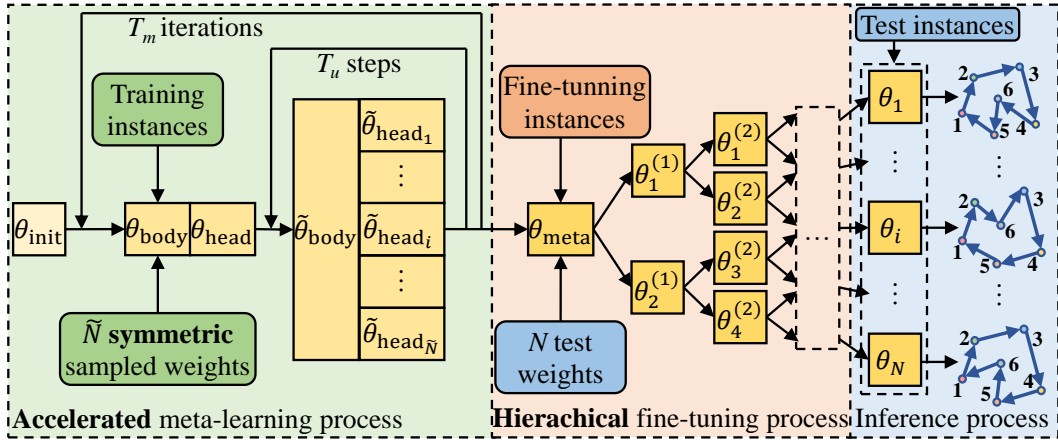

Figure 1: The overall framework of EMNH.

DRL-based neural heuristics [8–13] have also been investigated for MOCOPs. While bypassing the handcrafted rules, these neural heuristics adopt an end-to-end paradigm to construct solutions without iterative search. Benefiting from a large amount of data (i.e., problem instances), a well-trained deep model allows the neural heuristic to automatically extract informative and expressive features for decision making and generalize to unseen instances as well.

Although demonstrating promise for solving MOCOPs, those neural heuristics still struggle to achieve high learning efficiency and solution quality. Particularly, in this line of works, the early attempts [8–11] always train multiple deep models, i.e., one for each preference (weight) combination, making them less practical. While the well-known preference-conditioned multi-objective combinatorial optimization (PMOCO) [12] realized a unified deep model by introducing a huge hypernetwork, it leaves considerable gaps in terms of solution quality. The recent Meta-DRL (MDRL) [13] has demonstrated the capability to enhance solution quality over existing state-of-the-art algorithms. However, it still faces challenges related to inefficient and unstable training procedures, as well as undesirable fine-tuning processes. This paper thereby proposes an efficient meta neural heuristic (EMNH) to further strengthen the learning efficiency and solution quality, the framework of which is illustrated in Figure 1. Following the meta-learning paradigm [13, 14], EMNH first trains a meta-model and then quickly fine-tunes it according to the weight (preference) vector to tackle the corresponding single-objective subproblems.

Our contributions are summarized as follows. (1) We propose an efficient meta-learning scheme to accelerate the training. Inspired by the feature reuse of meta-learning [15], we introduce a multi-task model composed of a parameter-shared *body* (all other layers of neural network except the last one) and respective task-related *heads* (the last layer of the network), rather than respective submodels, to train the meta-model, which is able to handle different tasks (subproblems) in parallel. (2) We design a scaled symmetric sampling method regarding the weight vectors to stabilize the training. In each meta-iteration, once a weight vector is sampled, its scaled symmetric weight vectors will also be engendered, which help avoid fluctuations of the parameter update for the meta-model, especially on problems with objectives of imbalanced domains. (3) We present a hierarchical fine-tuning method to efficiently cope with all the subproblems. Since it is redundant to derive a specific submodel for each subproblem in the early tuning process, our meta-model can be gradually fine-tuned to act as specialized submodels from low level to high level, which takes much fewer fine-tuning steps in total. Moreover, experimental results on three classic MOCOPs confirmed the effectiveness of our designs.

## 2 Related works

**Exact and heuristic methods for MOCOPs.** Over the past decades, MOCOPs have been attracting increasing attention from the computational intelligence community, and tremendous works have been proposed. Generally, exact methods [2] are able to gain the accurate Pareto-optimal solutions, but their computational time may exponentially grow. Heuristic methods such as multi-objective

evolutionary algorithms (MOEAs) are popular in practice, where the representative paradigms for the general multi-objective optimization include NSGA-II [16] and MOEA/D [17]. Among them, a number of MOEAs outfitted with local search [18–20] are specifically designed for MOCOPs, which iteratively perform a search in the solution space to find approximate Pareto-optimal solutions.

**Neural heuristics for COPs.** As the early attempts, neural *construction* heuristics based on deep models [21–23] were proposed to learn to directly construct solutions for (single-objective) COPs. Among them, the classic *Attention Model* (AM) [24] was proposed inspired by *Transformer* architecture [25], and known as a milestone for solving vehicle routing problems. A number of subsequent works [26–30] were developed based on AM, including the popular policy optimization with multiple optima (POMO) [26], which leveraged the solution symmetries and significantly enhanced the performance over AM. Besides, graph neural networks were also employed to learn graphic embeddings for solving COPs [31–34]. Different from the above neural construction heuristics, neural *improvement* heuristics [35–39] were proposed to assist iteration-based methods to refine an initial solution.

**Neural heuristics for MOCOPs.** MOCOPs seek for a set of trade-off solutions, which could be obtained by solving a series of single-objective subproblems. According to the number of trained models, neural heuristics could be roughly classified into multi-model and single-model ones. The former adopts multiple networks to tackle respective subproblems, where the set of networks could also be collaboratively trained with a neighborhood-based parameter-transfer strategy [8, 9], or evolved with MOEA [10, 11]. Differently, the latter usually exploits a unified network to tackle all the subproblems, which is more flexible and practical. For example, taking the weight (preference) vector of a subproblem as the input, PMOCO introduces a hypernetwork to learn the decoder parameters [12]. However, the hypernetwork may cause extra complexity to the original deep reinforcement learning model, rendering it less effective to learn more accurate mapping from the preference to the optimal solution of the corresponding subproblem. By contrast, MDRL [13] leverages the meta-learning to train a deep reinforcement learning model that could be fine-tuned for various subproblems. However, its learning efficiency is far from optimal, especially given the slow and unstable training due to sequential learning of randomly sampled tasks, and inefficient fine-tuning due to redundant updates.

## 3 Preliminary

### 3.1 MOCOP

In general, an MOCOP could be defined as

$$\min_{x \in \mathcal{X}} \boldsymbol{f}(x) = (f_1(x), f_2(x), \ldots, f_M(x)), \tag{1}$$

where $\boldsymbol{f}(x)$ is an $M$-objective vector, $\mathcal{X}$ is a discrete decision space, and the objectives might be conflicted. The Pareto-optimal solutions concerned by decision makers are defined as follows.

**Definition 1 (Pareto dominance).** Let $u, v \in \mathcal{X}$, $u$ is said to dominate $v$ ($u \prec v$) if and only if $f_i(u) \leq f_i(v), \forall i \in \{1, \ldots, M\}$ and $\exists j \in \{1, \ldots, M\}, f_j(u) < f_j(v)$.

**Definition 2 (Pareto optimality).** A solution $x^* \in \mathcal{X}$ is Pareto-optimal if $x^*$ is not dominated by any other solution in $\mathcal{X}$, i.e., $\nexists x' \in \mathcal{X}$ such that $x' \prec x^*$. The set of all Pareto optimal solutions $\mathcal{P} = \{x^* \in \mathcal{X} \mid \nexists x' \in \mathcal{X} : x' \prec x^*\}$ is called the Pareto set. The *image* of Pareto set in the objective space, i.e., $\mathcal{PF} = \{\boldsymbol{f}(x) \mid x \in \mathcal{P}\}$ is called the Pareto front.

### 3.2 Decomposition

The decomposition strategy is always applied for solving MOCOPs [17], since it is straightforward yet effective to guide the search towards prescribed directions. Specifically, MOCOPs can be decomposed into $N$ scalarized single-objective subproblem $g(x|\boldsymbol{\lambda})$, where the weight vector $\boldsymbol{\lambda} \in \mathcal{R}^M$ satisfies $\lambda_m \geq 0$ and $\sum_{m=1}^M \lambda_m = 1$. Then, $\mathcal{PF}$ is approximated by solving the subproblems systematically.

Regarding the decomposition strategy, the weighted sum (WS) and Tchebycheff [40] are commonly used. As a simple representative, WS considers the linear combination of $M$ objectives as follows,

$$\min_{x \in \mathcal{X}} g_{\text{ws}}(x|\boldsymbol{\lambda}) = \sum_{m=1}^M \lambda_m f_m(x). \tag{2}$$

Given the weight vector $\boldsymbol{\lambda}$, the corresponding single-objective subproblem could be cast as a sequential decision problem and solved through DRL. In particular, a solution is represented as a sequence $\boldsymbol{\pi} = \{\pi_1, \ldots, \pi_T\}$ with length $T$, and a stochastic policy for yielding solution $\boldsymbol{\pi}$ from instance $s$ is calculated as $P(\boldsymbol{\pi}|s) = \prod_{t=1}^{T} P_{\boldsymbol{\theta}}(\pi_t|\boldsymbol{\pi}_{1:t-1}, s)$, where $P_{\boldsymbol{\theta}}(\pi_t|\boldsymbol{\pi}_{1:t-1}, s)$ is the node selection probability parameterized by $\boldsymbol{\theta}$ (the subscript of $\boldsymbol{\theta}_i$, $\boldsymbol{\pi}_i$, $P_i$ for subproblem $i$ is omitted for readability).

### 3.3 Meta-learning

Meta-learning [14, 41] aims to train a model that can learn to tackle various tasks (i.e., subproblems) via fine-tuning. Naturally, a meta-model can be trained to quickly adapt to $N$ corresponding subproblems given $N$ weight vectors. Generally, meta-learning comprises three processes for MOCOPs. In the meta-learning process, a model $\boldsymbol{\theta}$ is trained by repeatedly sampling tasks from the whole task space. In the fine-tuning process, $N$ submodels $\boldsymbol{\theta}_1, \ldots, \boldsymbol{\theta}_N$ based on the given weight vectors are fine-tuned from $\boldsymbol{\theta}$ using fine-tuning instances (also called query set). In the inference process, $N$ subproblems of an instance are solved using the submodels to approximate the Pareto set.

## 4 Methodology

Our *efficient meta neural heuristic* (EMNH) includes an accelerated and stabilized meta-learning process, a hierarchical fine-tuning process, and an inference process, as depicted in Figure 1. EMNH is generic, where we adopt the first-order gradient-based *Reptile* [14] as the backbone of meta-learning, and employ the popular neural solver POMO [26] as the base model. In our meta-learning process, the model $\boldsymbol{\theta}$ is trained with $T_m$ meta-iterations, where a multi-task model is exploited to speed up the training and a scaled symmetric sampling method is designed to stabilize the training. In our fine-tuning process, the meta-model $\boldsymbol{\theta}$ is hierarchically and quickly fine-tuned with a few gradient updates to solve the corresponding subproblems. The details for each design are presented below.

### 4.1 Accelerated training

In the accelerated training process, as shown in Algorithm 1, the meta-model $\boldsymbol{\theta}$ is trained with $T_m$ meta-iterations. In each iteration, the model needs to be optimized by learning $\tilde{N}$ sampled tasks (subproblems) with $T_u$-step gradient updates. Inspired by the feature reuse of meta-learning [15], it is unnecessary and expensive to serially update the meta-model through the $\tilde{N}$ respective submodels of the same architecture. On the other hand, it is reasonable to assume that only the *head* $\boldsymbol{\theta}_{\text{head}}$ is specified for a task, while the *body* $\boldsymbol{\theta}_{\text{body}}$ can be reused for all tasks. To realize such a lightweight meta-learning scheme, we further introduce a multi-task model $\tilde{\boldsymbol{\theta}}$ which is composed of a shared body and $\tilde{N}$ respective heads to learn the $\tilde{N}$ tasks in parallel.

Specifically, since our EMNH adopts the encoder-decoder structured POMO as the base model, as depicted in Figure 2(a), 1) $\boldsymbol{\theta}_{\text{head}}$ could be defined as the decoder head, i.e., $W^K \in \mathcal{R}^{d \times d}$ in the last single-head attention layer, where $d$ is empirically set to 128 [26]; 2) $\boldsymbol{\theta}_{\text{body}}$ could be composed of the whole encoder $\boldsymbol{\theta}_{\text{en}}$ and the decoder body $\boldsymbol{\theta}_{\text{de-body}}$. For a problem instance with $n$ nodes, the node embeddings $\boldsymbol{h}_1, \ldots, \boldsymbol{h}_n \in \mathcal{R}^d$ are computed by $\boldsymbol{\theta}_{\text{en}}$ at the encoding step. At each decoding step, the *query* $\boldsymbol{q}_c \in \mathcal{R}^d$ is first computed by $\boldsymbol{\theta}_{\text{de-body}}$ using the node embeddings and problem-specific *context* embedding $\boldsymbol{h}_c$. Then, the last single-head attention layer computes the probability of node selection $P_{\boldsymbol{\theta}}(\boldsymbol{\pi}|s)$ using $\boldsymbol{q}_c$ and the *key* $\boldsymbol{k}_1, \ldots, \boldsymbol{k}_n \in \mathcal{R}^d$, where $\boldsymbol{k}_{i'}$ for node $i'$ is computed by $\boldsymbol{\theta}_{\text{head}}$, i.e., $\boldsymbol{k}_{i'} = W^K \boldsymbol{h}_{i'}$. More details are presented in Appendix A.

As demonstrated in Figure 2(b), for natural accommodation, the multi-task model $\tilde{\boldsymbol{\theta}}$ consists of $\tilde{\boldsymbol{\theta}}_{\text{body}}$ and $\tilde{\boldsymbol{\theta}}_{\text{head}_1}, \ldots, \tilde{\boldsymbol{\theta}}_{\text{head}_{\tilde{N}}}$, where $\tilde{\boldsymbol{\theta}}_{\text{body}}$ and $\tilde{\boldsymbol{\theta}}_{\text{head}_i}$ have the same architecture as $\boldsymbol{\theta}_{\text{body}}$ and $\boldsymbol{\theta}_{\text{head}}$, respectively. Note that $\tilde{\boldsymbol{\theta}}_{\text{head}_i}$ is individually updated for subproblem $i$, while $\tilde{\boldsymbol{\theta}}_{\text{body}}$ is shared for $\tilde{N}$ tasks. Concretely, since the subproblems are captured by different weight vectors but the same (or similar) node features, the shared node embeddings are computed by $\tilde{\boldsymbol{\theta}}_{\text{en}}$. At each decoding step, for subproblem $i$, the *query* $\boldsymbol{q}_{c,i}$ is first computed by $\tilde{\boldsymbol{\theta}}_{\text{de-body}}$ using the node embeddings and *context* embedding $\boldsymbol{h}_{c,i}$. Then, the *key* is computed as $\boldsymbol{k}_{i',i} = W_i^K \boldsymbol{h}_{i'}$ for node $i'$. Finally, the single-head attention layer computes the probability of node selection $P_{\tilde{\boldsymbol{\theta}}_i}(\boldsymbol{\pi}|s)$ for subproblem $i$ using $\boldsymbol{q}_{c,i}$ and $\boldsymbol{k}_{1,i}, \ldots, \boldsymbol{k}_{n,i}$. In practice, $W_1^K, \ldots, W_{\tilde{N}}^K$ are concatenated to tackle the $\tilde{N}$ tasks in parallel.

---

**Algorithm 1** Accelerated training process

---

**Input:** initialized meta-model $\boldsymbol{\theta}$, number of symmetric sampled weight vectors $\tilde{N}$, initialized multi-task model $\tilde{\boldsymbol{\theta}}$, initial meta-learning rate $\epsilon_0$, number of meta-iterations $T_m$, number of update steps of the multi-task model $T_u$, batch size $B$, problem size $n$

1:  $\epsilon \leftarrow \epsilon_0$
2: **for** $t_m = 1$ to $T_m$ **do**
3:    $\boldsymbol{\lambda}_i$ is obtained by the scaled symmetric sampling method, $\quad \forall i \in \{1, \dots, \tilde{N}\}$
4:    $\tilde{\boldsymbol{\theta}}_{\text{body}} \leftarrow \boldsymbol{\theta}_{\text{body}}$
5:    $\tilde{\boldsymbol{\theta}}_{\text{head}_i} \leftarrow \boldsymbol{\theta}_{\text{head}}, \quad \forall i$
6:    **for** $t_u = 1$ to $T_u$ **do**
7:      $s_j \sim \textbf{SampleInstance}(\mathcal{S}), \quad \forall j \in \{1, \dots, B\}$
8:      $\{\boldsymbol{\pi}^k|s_j, \boldsymbol{\lambda}_i\} \sim \textbf{SampleRollout}(P_{\tilde{\boldsymbol{\theta}}i}(\cdot|s_j)), \quad \forall k \in \{1, \dots, n\}, \forall i, \forall j$
9:      $b_{ij} \leftarrow \frac{1}{n}\sum_{k=1}^{n} g(\boldsymbol{\pi}^k|s_j, \boldsymbol{\lambda}_i)$
10:     $\nabla\mathcal{L}(\tilde{\boldsymbol{\theta}}) \leftarrow \frac{1}{\tilde{N}Bn}\sum_{i=1}^{\tilde{N}}\sum_{j=1}^{B}\sum_{k=1}^{n}[(g_{ij}^k - b_{ij})\nabla\log P_{\tilde{\boldsymbol{\theta}}i}(\boldsymbol{\pi}^k|s_j)]$
11:     $\tilde{\boldsymbol{\theta}} \leftarrow \textbf{Adam}(\tilde{\boldsymbol{\theta}}, \nabla\mathcal{L}(\tilde{\boldsymbol{\theta}}))$
12:    **end for**
13:    $\boldsymbol{\theta}_{\text{body}} \leftarrow \tilde{\boldsymbol{\theta}}_{\text{body}}$
14:    $\boldsymbol{\theta}_{\text{head}} \leftarrow \boldsymbol{\theta}_{\text{head}} + \epsilon\left(\frac{1}{\tilde{N}}\sum_{i=1}^{\tilde{N}}\tilde{\boldsymbol{\theta}}_{\text{head}_i} - \boldsymbol{\theta}_{\text{head}}\right)$
15:    $\epsilon \leftarrow \epsilon - \epsilon_0/T_m$
16: **end for**
**Output:** The trained meta-model $\boldsymbol{\theta}$

---

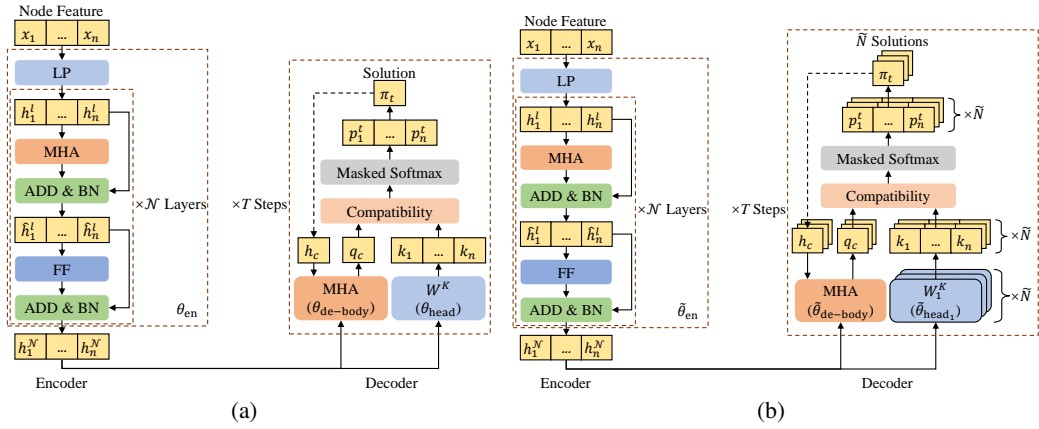

Figure 2: Model architectures. (a) Base model (POMO). (b) Multi-task model.

We extend REINFORCE [42] to train the multi-task model $\tilde{\boldsymbol{\theta}}$, including $\tilde{\boldsymbol{\theta}}_{\text{body}}$ for all tasks and $\tilde{\boldsymbol{\theta}}_{\text{head}_i}$ for task $i$. Specifically, $\tilde{\boldsymbol{\theta}}$ is optimized by gradient descent with the multi-task loss $\nabla\mathcal{L} = \frac{1}{\tilde{N}}\sum_{i=1}^{\tilde{N}}\nabla\mathcal{L}_i$, where $\mathcal{L}_i(\tilde{\boldsymbol{\theta}}|s)$ is the $i$-th task loss and its gradient is estimated as follows,

$$\nabla\mathcal{L}_i(\tilde{\boldsymbol{\theta}}|s) = \mathbf{E}_{P_{\tilde{\boldsymbol{\theta}}i}(\boldsymbol{\pi}|s)}[(g(\boldsymbol{\pi}|s, \boldsymbol{\lambda}_i) - b_i(s))\nabla\log P_{\tilde{\boldsymbol{\theta}}i}(\boldsymbol{\pi}|s)], \tag{3}$$

where $b_i(s) = \frac{1}{n}\sum_{k=1}^{n} g(\boldsymbol{\pi}^k|s, \boldsymbol{\lambda}_i)$ is a baseline to reduce the gradient variance; $\boldsymbol{\pi}^1, \dots, \boldsymbol{\pi}^n$ are solutions produced by POMO with $n$ different starting nodes. In practice, $\nabla\mathcal{L}$ is approximated using Monte Carlo sampling, and computed by a batch with $B$ instances as follows,

$$\nabla\mathcal{L}(\tilde{\boldsymbol{\theta}}) \approx \frac{1}{\tilde{N}Bn}\sum_{i=1}^{\tilde{N}}\sum_{j=1}^{B}\sum_{k=1}^{n}[(g_{ij}^k - b_{ij})\nabla\log P_{\tilde{\boldsymbol{\theta}}i}(\boldsymbol{\pi}^k|s_j)], \tag{4}$$

where $g_{ij}^k = g(\boldsymbol{\pi}^k|s_j, \boldsymbol{\lambda}_i)$ and $b_{ij} = b_i(s_j)$. $\tilde{\boldsymbol{\theta}}$ is then optimized with $T_u$ steps using Adam [43].

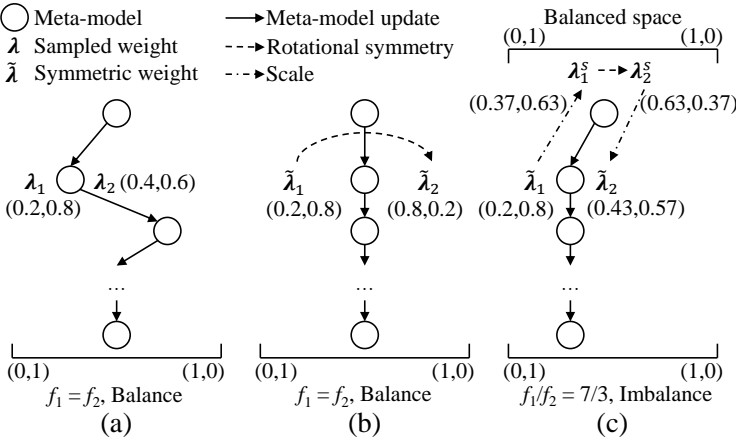

Figure 3: (a) Random sampling. (b) Symmetric sampling. (c) Scaled symmetric sampling.

## 4.2 Stabilized training

In each meta-iteration, the deviation of $\tilde{N}$ (usually a small number) randomly sampled weight vectors may cause fluctuations to the parameter update of the meta-model, thereby leading to unstable training performance. For example, as shown in Figure 3(a), the biased samples of weight vectors, i.e., $\lambda_1$ and $\lambda_2$, make the meta-model tend to favor the second objective more in this iteration.

To address this issue, we first design a symmetric sampling method for the case of balanced objective domain, as illustrated in Figure 3(b). Once a weight vector $\tilde{\lambda}_1$ is sampled, its $M$-1 rotational symmetric weight vectors $\tilde{\lambda}_2, \ldots, \tilde{\lambda}_M$ are calculated and used to reduce the bias as follows,

$$\tilde{\lambda}_{i,m} = \begin{cases} \tilde{\lambda}_{i-1,M}, & m = 1 \\ \tilde{\lambda}_{i-1,m-1}, & \text{otherwise} \end{cases} . \tag{5}$$

For the case of imbalanced objective domains, we further design a scaled symmetric sampling method. In particular, the $m$-th weight $\lambda_m$ is scaled as $\lambda_m/f_m^*$, where $f_m^*$ is the ideal objective value but difficult to be calculated beforehand. Thus, $f_m^*$ is replaced with $f_m'$, which is dynamically estimated using the meta-model on a validation dataset with $\lambda = (1/M, \ldots, 1/M)$ during training.

Concretely, for a sampled $\tilde{\lambda}_1$, it is first multiplied by $f'$ to scale objectives into the balanced domains. Then, its $M$-1 rotational symmetric weight vectors are obtained by Eq. (5). Finally, they are divided by $f'$ to scale back to the original domain to attain the genuine rotational symmetric weight vectors. An example for $\tilde{\lambda}_1 = (0.2, 0.8)$ and its scaled symmetric $\tilde{\lambda}_2 = (0.43, 0.57)$ are demonstrated in Figure 3(c). In summary, the scaled symmetric $\tilde{\lambda}_2, \ldots, \tilde{\lambda}_M$ for a given $\tilde{\lambda}_1$ are calculated as follows,

$$\lambda_{i,m}^s = \begin{cases} \tilde{\lambda}_{i-1,M} \times f_M'/f_m', & m = 1 \\ \tilde{\lambda}_{i-1,m-1} \times f_{m-1}'/f_m', & \text{otherwise} \end{cases} . \tag{6}$$

Then, the vectors are normalized as $\tilde{\lambda}_{i,m} = \frac{\lambda_{i,m}^s}{\sum_{m=1}^M \lambda_{i,m}^s}$. The pseudo code of the scaled symmetric sampling method for the weight vectors is presented in Appendix B.

## 4.3 Hierarchical fine-tuning

Once the meta-model is trained, it can be fine-tuned with a number of gradient updates to derive customized submodels for given weight vectors. However, it still might be expensive for MOCOPs since numerous submodels need to be individually fine-tuned from the meta-model. From a systematic perspective, the early coarse-tuning processes in which the parameters of neighboring submodels might be close, could be merged. In this sense, we propose a hierarchical method to efficiently fine-tune the numerous subproblems from the lowest to the highest level, as illustrated in Figure 1.

We suggest an $L$-level and $a$-section hierarchy to gradually produce more specific submodels. Particularly, all $N^{(l)}$ weight vectors in level $l$ are defined as the centers of their corresponding subspaces that uniformly divide the whole weight vector space based on the Das and Dennis [44] method. $N^{(l)}$ submodels are then fine-tuned to derive $N^{(l+1)} = aN^{(l)}$ submodels with $K^{(l+1)}$ fine-tuning steps in level $l$+1. Finally, $N^{(L-1)}$ submodels are fine-tuned to derive $N^{(L)} = N$ submodels, which are used to solve the $N$ given subproblems. More details of this fine-tuning are presented in Appendix C.

For the $L$-level and $a$-section hierarchy with $a^L = N$ and $K^{(1)} = \cdots = K^{(L)} = K$, the total fine-tuning step is $T_f^h = \sum_{l=1}^{L} Ka^l = Ka(a^L - 1)/(a - 1)$. For the vanilla process, each submodel needs $KL$ fine-tuning steps to achieve comparable performance, thus $T_f^v = KLN = KLa^L$ steps in total. Hence, the hierarchical fine-tuning only needs $T_f^h/T_f^v \approx 1/L$ of the vanilla fine-tuning steps.

## 5 Experimental results

### 5.1 Experimental settings

We conduct computational experiments to evaluate the proposed method on the multi-objective traveling salesman problem (MOTSP), multi-objective capacitated vehicle routing problem (MOCVRP), and multi-objective knapsack problem (MOKP). Following the convention in [26, 12], we consider the instances of different sizes $n$=20/50/100 for MOTSP/MOCVRP and $n$=50/100/200 for MOKP. All experiments are run on a PC with an Intel Xeon 4216 CPU and an RTX 3090 GPU.

**Problems.** Four classes of MOTSP [45] are considered, including bi-objective TSP type 1&2 (Bi-TSP-1, Bi-TSP-2) and tri-objective TSP type 1&2 (Tri-TSP-1, Tri-TSP-2). For the $M$-objective TSP type 1, node $i$ has $M$ 2D coordinates $\{\boldsymbol{x}_i^1, \ldots, \boldsymbol{x}_i^M\}$, where the $m$-th cost between node $i$ and $j$ is defined as the Euclidean distance $c_{ij}^m = \|\boldsymbol{x}_i^m - \boldsymbol{x}_j^m\|_2$. For the $M$-objective TSP type 2, a node contains $M$-1 2D coordinates together with the altitude, so the $M$-th cost is defined as the altitude variance. For the bi-objective CVRP (Bi-CVRP) [46], two conflicting objectives with imbalanced domains, i.e., the total traveling distance and the traveling distance of the longest route (also called makespan), are considered. For the bi-objective KP (Bi-KP) [47], item $i$ has a single weight and two distinct values. More details of these MOCOPs are presented in Appendix D.

**Hyper-parameters.** The meta-learning rate $\epsilon$ is linearly annealed to 0 from $\epsilon_0 = 1$ initially. A constant learning rate of the Adam optimizer is set to $10^{-4}$. We set $B = 64$, $T_m = 3000$, $T_u = 100$, and $\tilde{N} = M$. The $N$ weight vectors in $\mathcal{PF}$ construction are produced according to [44], where $N = C_{H+M-1}^{M-1}$. $H$ is set to 100 ($N = 101$) and 13 ($N = 105$) for $M = 2$ and $M = 3$, respectively. WS scalarization is adopted. In the hierarchy of fine-tuning, the whole weight space is uniformly divided by the Das and Dennis [44] method with $H^{(l)} = 2^l$, i.e., level $l$ has $2^l$ or $4^l$ subspaces and $L = 7$ or $L = 4$ for $M = 2$ or $M = 3$. The number of fine-tuning steps at each level is $K^{(l)} = K = 20$ or $K^{(l)} = K = 25$ for $M = 2$ or $M = 3$.

**Baselines.** Two kinds of strong baselines are introduced, and all of them adopt WS (weighted sum) scalarization for fair comparisons. (1) The state-of-the-art neural heuristics, including **MDRL** [13], **PMOCO** [12], and DRL-based multiobjective optimization algorithm (**DRL-MOA**) [8]. Same as EMNH, all these neural heuristics adopt POMO as the base model for single-objective subproblems. (2) The state-of-the-art traditional heuristics, including **PPLS/D-C** [20], **WS-LKH**, and **WS-DP**. PPLS/D-C is a local-search-based MOEA proposed for MOCOPs, where a 2-opt heuristic is used for MOTSP and MOCVRP, and a greedy transformation heuristic [48] is used for MOKP, running with 200 iterations for all the three problems. WS-LKH and WS-DP are based on the state-of-the-art heuristics for decomposed single-objective subproblems, i.e., LKH [49, 50] for MOTSP and dynamic programming (DP) for MOKP, respectively. Our code is publicly available[2].

**Metrics.** Our EMNH is evaluated in terms of solution quality and learning efficiency. Solution quality is mainly measured by hypervolume (HV) [51], where a higher HV means a better solution set (the definition is presented in Appendix E). Learning efficiency mainly refers to the training and fine-tuning efficiency, which are measured by training time for the same amount of training instances and solution quality with the same total fine-tuning steps (see Appendix F), respectively.

---

[2]https://github.com/bill-cjb/EMNH

Table 1: Results on 200 random instances for MOCOPs with balanced objective domains.

| Method | Bi-TSP-1 ($n$=20) | | | Bi-TSP-1 ($n$=50) | | | Bi-TSP-1 ($n$=100) | | |
| | HV↑ | Gap↓ | Time | HV↑ | Gap↓ | Time | HV↑ | Gap↓ | Time |
| --- | --- | --- | --- | --- | --- | --- | --- | --- | --- |
| WS-LKH | 0.6270 | 0.02% | 10m | **0.6415** | **-0.11%** | 1.8h | **0.7090** | **-0.95%** | 6.0h |
| PPLS/D-C | 0.6256 | 0.24% | 26m | 0.6282 | 1.97% | 2.8h | 0.6844 | 2.55% | 11h |
| DRL-MOA | 0.6257 | 0.22% | 6s | 0.6360 | 0.75% | 9s | 0.6970 | 0.75% | 21s |
| PMOCO | 0.6259 | 0.19% | 6s | 0.6351 | 0.89% | 10s | 0.6957 | 0.94% | 19s |
| MDRL | **0.6271** | **0.00%** | 5s | 0.6364 | 0.69% | 9s | 0.6969 | 0.77% | 17s |
| EMNH | **0.6271** | **0.00%** | 5s | 0.6364 | 0.69% | 9s | 0.6969 | 0.77% | 16s |
| PMOCO-Aug | 0.6270 | 0.02% | 45s | 0.6395 | 0.20% | 2.3m | 0.7016 | 0.10% | 15m |
| MDRL-Aug | **0.6271** | **0.00%** | 33s | 0.6408 | 0.00% | 1.7m | 0.7022 | 0.01% | 14m |
| EMNH-Aug | **0.6271** | **0.00%** | 33s | 0.6408 | 0.00% | 1.7m | 0.7023 | 0.00% | 14m |

| Method | Bi-KP ($n$=50) | | | Bi-KP ($n$=100) | | | Bi-KP ($n$=200) | | |
| | HV↑ | Gap↓ | Time | HV↑ | Gap↓ | Time | HV↑ | Gap↓ | Time |
| --- | --- | --- | --- | --- | --- | --- | --- | --- | --- |
| WS-DP | **0.3561** | **0.00%** | 22m | 0.4532 | 0.07% | 2.0h | 0.3601 | 0.06% | 5.8h |
| PPLS/D-C | 0.3528 | 0.93% | 18m | 0.4480 | 1.21% | 47m | 0.3541 | 1.72% | 1.5h |
| DRL-MOA | 0.3559 | 0.06% | 9s | 0.4531 | 0.09% | 18s | 0.3601 | 0.06% | 1.0m |
| PMOCO | 0.3552 | 0.25% | 6s | 0.4523 | 0.26% | 22s | 0.3595 | 0.22% | 1.3m |
| MDRL | 0.3530 | 0.87% | 6s | 0.4532 | 0.07% | 21s | 0.3601 | 0.06% | 1.2m |
| EMNH | **0.3561** | **0.00%** | 6s | **0.4535** | **0.00%** | 21s | **0.3603** | **0.00%** | 1.2m |

| Method | Tri-TSP-1 ($n$=20) | | | Tri-TSP-1 ($n$=50) | | | Tri-TSP-1 ($n$=100) | | |
| | HV↑ | Gap↓ | Time | HV↑ | Gap↓ | Time | HV↑ | Gap↓ | Time |
| --- | --- | --- | --- | --- | --- | --- | --- | --- | --- |
| WS-LKH | **0.4712** | **0.00%** | 12m | **0.4440** | **-0.50%** | 1.9h | **0.5076** | **-2.07%** | 6.6h |
| PPLS/D-C | 0.4698 | 0.30% | 1.4h | 0.4174 | 5.52% | 3.9h | 0.4376 | 12.00% | 14h |
| DRL-MOA | 0.4699 | 0.28% | 6s | 0.4303 | 2.60% | 9s | 0.4806 | 3.36% | 19s |
| PMOCO | 0.4693 | 0.40% | 5s | 0.4315 | 2.33% | 8s | 0.4858 | 2.31% | 18s |
| MDRL | 0.4699 | 0.28% | 5s | 0.4317 | 2.29% | 9s | 0.4852 | 2.43% | 16s |
| EMNH | 0.4699 | 0.28% | 5s | 0.4324 | 2.13% | 9s | 0.4866 | 2.15% | 16s |
| PMOCO-Aug | **0.4712** | **0.00%** | 3.2m | 0.4409 | 0.20% | 28m | 0.4956 | 0.34% | 1.7h |
| MDRL-Aug | **0.4712** | **0.00%** | 2.6m | 0.4408 | 0.23% | 25m | 0.4958 | 0.30% | 1.7h |
| EMNH-Aug | **0.4712** | **0.00%** | 2.6m | 0.4418 | 0.00% | 25m | 0.4973 | 0.00% | 1.7h |

## 5.2 Solution quality

The results for MOTSP, MOCVRP, and MOKP are recorded in Tables 1 and 2, including the average HV, gap, and total running time for solving 200 random test instances. To further show the significant differences between the results, a Wilcoxon rank-sum test at 1% significance level is adopted. The best result and the one without statistical significance to it are highlighted as **bold**. The second-best result and the one without statistical significance to it are highlighted as underline. The method with "-Aug" represents the inference results using instance augmentation [12] (see Appendix G).

**Optimality gap.** The gaps of HV with respect to EMNH-Aug are reported for all methods. According to the results in Tables 1 and 2, EMNH outperforms other neural heuristics without instance augmentation on all problems. When equipped with instance augmentation, EMNH-Aug further improves the solution, and performs superior to most of the baselines, while slightly inferior to WS-LKH on MOTSP with $n$=50 and $n$=100. However, as iteration-based methods, WS-LKH and MOEAs take quite long running time on MOCOPs. For the neural heuristics, DRL-MOA is less agile since it trains multiple fixed models for a priori weight vectors. For new weight vectors concerned by decision makers, EMNH, MDRL, and PMOCO are able to efficiently produce high-quality trade-off solutions, where EMNH achieves the smallest gap.

**Imbalanced objective domains.** As demonstrated in Table 2, the gaps between EMNH and PMOCO or MDRL are further increased on the problems with imbalanced objective domains, i.e., Bi-CVRP, Bi-TSP-2, and Tri-TSP-2. EMNH exhibits more competitive performance on these problems, revealing

Table 2: Results on 200 random instances for MOCOPs with imbalanced objective domains.

| Method | Bi-CVRP ($n$=20) | | | Bi-CVRP ($n$=50) | | | Bi-CVRP ($n$=100) | | |
|---|---|---|---|---|---|---|---|---|---|
| | HV↑ | Gap↓ | Time | HV↑ | Gap↓ | Time | HV↑ | Gap↓ | Time |
| PPLS/D-C | 0.4287 | 0.35% | 1.6h | 0.4007 | 2.41% | 9.7h | 0.3946 | 3.26% | 38h |
| DRL-MOA | 0.4287 | 0.35% | 10s | 0.4076 | 0.73% | 12s | 0.4055 | 0.59% | 33s |
| PMOCO | 0.4267 | 0.81% | 7s | 0.4036 | 1.70% | 12s | 0.3913 | 4.07% | 32s |
| MDRL | 0.4291 | 0.26% | 8s | 0.4082 | 0.58% | 13s | 0.4056 | 0.56% | 32s |
| EMNH | 0.4299 | 0.07% | 8s | 0.4098 | 0.19% | 12s | 0.4072 | 0.17% | 32s |
| PMOCO-Aug | 0.4294 | 0.19% | 13s | 0.4080 | 0.63% | 36s | 0.3969 | 2.70% | 2.7m |
| MDRL-Aug | 0.4294 | 0.19% | 11s | 0.4092 | 0.34% | 36s | 0.4072 | 0.17% | 2.8m |
| EMNH-Aug | **0.4302** | **0.00%** | 11s | **0.4106** | **0.00%** | 35s | **0.4079** | **0.00%** | 2.8m |

| Method | Bi-TSP-2 ($n$=20) | | | Bi-TSP-2 ($n$=50) | | | Bi-TSP-2 ($n$=100) | | |
|---|---|---|---|---|---|---|---|---|---|
| | HV↑ | Gap↓ | Time | HV↑ | Gap↓ | Time | HV↑ | Gap↓ | Time |
| WS-LKH | 0.6660 | 0.13% | 11m | **0.7390** | **-0.07%** | 1.8h | **0.8055** | **-0.74%** | 6.1h |
| PPLS/D-C | 0.6662 | 0.10% | 27m | 0.7300 | 1.15% | 3.3h | 0.7859 | 1.71% | 10h |
| DRL-MOA | 0.6657 | 0.18% | 6s | 0.7359 | 0.35% | 8s | 0.7965 | 0.39% | 18s |
| PMOCO | 0.6590 | 1.18% | 5s | 0.7347 | 0.51% | 8s | 0.7944 | 0.65% | 17s |
| MDRL | **0.6669** | **0.00%** | 5s | 0.7361 | 0.32% | 9s | 0.7965 | 0.39% | 15s |
| EMNH | **0.6669** | **0.00%** | 5s | 0.7361 | 0.32% | 9s | 0.7965 | 0.39% | 15s |
| PMOCO-Aug | 0.6653 | 0.24% | 13s | 0.7375 | 0.14% | 33s | 0.7988 | 0.10% | 3.5m |
| MDRL-Aug | **0.6669** | **0.00%** | 8s | 0.7385 | 0.00% | 22s | 0.7996 | 0.00% | 2.8m |
| EMNH-Aug | **0.6669** | **0.00%** | 8s | 0.7385 | 0.00% | 22s | 0.7996 | 0.00% | 2.8m |

| Method | Tri-TSP-2 ($n$=20) | | | Tri-TSP-2 ($n$=50) | | | Tri-TSP-2 ($n$=100) | | |
|---|---|---|---|---|---|---|---|---|---|
| | HV↑ | Gap↓ | Time | HV↑ | Gap↓ | Time | HV↑ | Gap↓ | Time |
| WS-LKH | **0.5035** | **0.00%** | 13m | **0.5305** | **-0.49%** | 2.0h | **0.5996** | **-1.70%** | 6.6h |
| PPLS/D-C | **0.5035** | **0.00%** | 1.4h | 0.5045 | 4.42% | 4.1h | 0.5306 | 10.01% | 15h |
| DRL-MOA | 0.5019 | 0.32% | 6s | 0.5101 | 3.37% | 8s | 0.5488 | 6.92% | 19s |
| PMOCO | 0.5020 | 0.30% | 5s | 0.5176 | 1.95% | 8s | 0.5777 | 2.02% | 18s |
| MDRL | 0.5024 | 0.22% | 5s | 0.5183 | 1.82% | 9s | 0.5806 | 1.53% | 16s |
| EMNH | 0.5024 | 0.22% | 5s | 0.5205 | 1.40% | 9s | 0.5813 | 1.41% | 16s |
| PMOCO-Aug | **0.5035** | **0.00%** | 55s | 0.5258 | 0.40% | 6.1m | 0.5862 | 0.58% | 32m |
| MDRL-Aug | **0.5035** | **0.00%** | 37s | 0.5267 | 0.23% | 4.2m | 0.5886 | 0.17% | 30m |
| EMNH-Aug | **0.5035** | **0.00%** | 37s | 0.5279 | 0.00% | 4.2m | 0.5896 | 0.00% | 30m |

the effectiveness of the proposed scaled symmetric sampling method in tackling the imbalance of objective domains. We further equip PMOCO with the same sampling method, which actually also improved the solutions for those problems, but still performed inferior to EMNH (see Appendix H).

**Generalization ability.** We test the generalization ability of the model on 200 larger-scale random instances ($n$=150/200) and 3 commonly used MOTSP benchmark instances (KroAB100/150/200) in TSPLIB [52]. The zero-shot generalization performance of the model trained and fine-tuned both on the instances with $n$=100 is reported in Appendix I. The results suggest that EMNH has a superior generalization ability compared with MOEAs and other neural heuristics for larger problem sizes.

**Hyper-parameter study.** The results in Appendix J showed that the number of sampled weight vectors $\tilde{N} = M$ is a more desirable setting and WS is a simple yet effective scalarization method.

## 5.3 Learning efficiency

As verified above, EMNH is able to produce superior solutions to the state-of-the-art neural heuristics, especially demonstrating a significant advantage over PMOCO in terms of solution quality. Then we further show that EMNH also has favorable learning efficiency against MDRL.

**Training efficiency.** EMNH, MDRL, and PMOCO only train one model to tackle all subproblems, where EMNH and MDRL use the same amount of training instances, and PMOCO requires a few

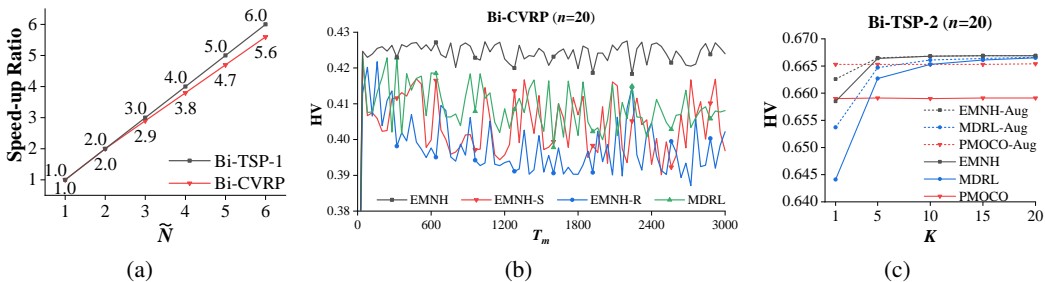

Figure 4: Learning efficiency. (a) Training efficiency. (b) Training stability. (c) Fine-tuning efficiency.

more training instances according to its original setting. DRL-MOA needs to train multiple submodels with more training instances. More details are presented in Appendix F. Figure 4(a) displays the speed-up ratio, i.e., the ratio of the training time of MDRL to EMNH. Due to the multi-task based training for (partial) architecture reuse, training time of EMNH is only about $1/\tilde{N}$ of that of MDRL.

**Training stability.** Figure 4(b) shows the stability of the training process, where EMNH-S and EMNH-R refer to EMNH with the symmetric sampling and the random sampling, respectively. EMNH achieves the stablest and best training performance, compared to EMNH-S, EMNH-R, and MDRL, as the proposed sampling method in EMNH considers symmetric weight vectors and (imbalanced) objective domains. More details are presented in Appendix F.

**Fine-tuning efficiency.** Figure 4(c) compares the hierarchical fine-tuning in EMNH with the vanilla fine-tuning method in MDRL, where the HV of EMNH with $K$ fine-tuning steps at each level and the HV of MDRL with approximately equal total fine-tuning steps are presented. The results demonstrate that EMNH attains higher fine-tuning efficiency than MDRL, i.e., larger gaps with smaller fine-tuning steps. Furthermore, PMOCO is also equipped with the hierarchical fine-tuning (for both versions) for a fair comparison. It is worth noting that our fine-tuning process is performed individually for each weight vector on dedicated fine-tuning instances. As can be seen, PMOCO could hardly get improved by fine-tuning for zero-shot inference on test instances, since it has already converged for the corresponding subproblems. This means that EMNH has favorable potential to derive more desirable submodels to tackle specific tasks. Notably, EMNH with a few fine-tuning steps (e.g., $K = 5$) outperforms PMOCO in most cases, as demonstrated in Appendix F.

## 6 Conclusion

This paper proposes an efficient meta neural heuristic (EMNH) for MOCOPs. Specifically, EMNH reduces training time, stabilizes the training process, and curtails total fine-tuning steps via (partial) architecture-shared multi-task learning, scaled symmetric sampling (of weight vectors), and hierarchical fine-tuning, respectively. The experimental results on MOTSP, MOCVRP, and MOKP verified the superiority of EMNH in learning efficiency and solution quality. A limitation is that our EMNH, as the same as other neural heuristics, can not guarantee obtaining the exact Pareto front. In the future, we will (1) extend EMNH to other MOCOPs with complex constraints; (2) investigate other advanced meta-learning algorithms and neural solvers as the base model to further improve the performance.

## Acknowledgments and disclosure of funding

This work is supported by the National Natural Science Foundation of China (62072483), and the Guangdong Basic and Applied Basic Research Foundation (2022A1515011690, 2021A1515012298).

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
