# Efficient Meta Neural Heuristic for Multi-Objective Combinatorial Optimization (Appendix)

## A  Model architecture

The architecture of the base model in meta-learning is the same as POMO [26], composed of an encoder and a decoder (see Figure 2(a)). For node features $\boldsymbol{x}_1, \ldots, \boldsymbol{x}_n$, the encoder first computes initial node embeddings $\boldsymbol{h}_1^0, \ldots, \boldsymbol{h}_n^0 \in \mathcal{R}^d$ ($d = 128$) by a linear projection (LP). The final node embeddings $\boldsymbol{h}_1^{\mathcal{N}}, \ldots, \boldsymbol{h}_n^{\mathcal{N}}$ are further computed by $\mathcal{N} = 6$ attention layers. Each attention layer is composed of a multi-head attention (MHA) with $M = 8$ heads and fully connected feed-forward (FF) sublayer. Each sublayer adds a skip-connection (ADD) and batch normalization (BN).

The decoder sequentially chooses a node according to a probability distribution produced by the node embeddings to construct a solution. The total decoding step $T$ is determined by the specific problem. At step $t$ in the decoding procedure, the *query* $\boldsymbol{q}_c \in \mathcal{R}^d$ is computed by an MHA layer using the node embeddings and problem-specific *context* embedding $\boldsymbol{h}_c$ (see Appendix D). The *key* $\boldsymbol{k}_1, \ldots, \boldsymbol{k}_n \in \mathcal{R}^d$ is computed by $\boldsymbol{k}_{i'} = W^K \boldsymbol{h}_{i'}$ for node $i'$. Then, in the final single-head attention layer, the *compatibility* $\boldsymbol{u}$ is computed by $\boldsymbol{q}_c$ and $\boldsymbol{k}_{i'}$ as follows,

$$
u_{i'} = \begin{cases} -\infty, & \text{node } i' \text{ is masked} \\ C \cdot \tanh(\frac{\boldsymbol{q}_c^T \boldsymbol{k}_{i'}}{\sqrt{d/M}}), & \text{otherwise} \end{cases} \tag{7}
$$

where $C = 10$ is adopted to clip the result. Finally, the probability distribution $P_{\boldsymbol{\theta}}(\boldsymbol{\pi}|s)$ to select node $i'$ is computed by the softmax function,

$$
P_{i'} = P_{\boldsymbol{\theta}}(\pi_t = i'|\boldsymbol{\pi}_{1:t-1}, s) = \frac{e^{u_{i'}}}{\sum_{j=1}^{n} e^{u_j}}. \tag{8}
$$

For the meta-model $\boldsymbol{\theta}$, $\boldsymbol{\theta}_{\text{head}}$ can be defined as $W^K$ for the final single-head attention layer, and $\boldsymbol{\theta}_{\text{body}}$ is composed of the whole encoder $\boldsymbol{\theta}_{\text{en}}$ and the decoder body $\boldsymbol{\theta}_{\text{de-body}}$.

In each meta-iteration, we adopt a multi-task model $\tilde{\boldsymbol{\theta}}$, as shown in Figure 2(b), to learn $\tilde{N}$ sampled tasks in parallel. The multi-task model $\tilde{\boldsymbol{\theta}}$ consists of $\tilde{\boldsymbol{\theta}}_{\text{body}}$ and $\tilde{\boldsymbol{\theta}}_{\text{head}_1}, \ldots, \tilde{\boldsymbol{\theta}}_{\text{head}_{\tilde{N}}}$, where $\tilde{\boldsymbol{\theta}}_{\text{body}}$ and $\tilde{\boldsymbol{\theta}}_{\text{head}_i}$ have the same architecture as $\boldsymbol{\theta}_{\text{body}}$ and $\boldsymbol{\theta}_{\text{head}}$, respectively. $\tilde{\boldsymbol{\theta}}_{\text{head}_i}$, i.e., $W_i^K$, is individually updated for subproblem $i$, while $\tilde{\boldsymbol{\theta}}_{\text{body}}$ is shared across $\tilde{N}$ tasks. Specifically, the shared node embeddings are first computed by $\tilde{\boldsymbol{\theta}}_{\text{en}}$. Then, at step $t$ in the decoding procedure, for subproblem $i$, the *query* $\boldsymbol{q}_{c,i}$ is computed using the node embeddings and *context* embedding $\boldsymbol{h}_{c,i}$. The *key* $\boldsymbol{k}_{1,i}, \ldots, \boldsymbol{k}_{n,i}$ is computed by $\boldsymbol{k}_{i',i} = W_i^K \boldsymbol{h}_{i'}$. The *compatibility* $\boldsymbol{u}_i$ is computed as follows,

$$
u_{i',i} = \begin{cases} -\infty, & \text{node } i' \text{ is masked} \\ C \cdot \tanh(\frac{\boldsymbol{q}_{c,i}^T \boldsymbol{k}_{i',i}}{\sqrt{d/M}}), & \text{otherwise} \end{cases} \tag{9}
$$

Finally, the probability $P_{\tilde{\boldsymbol{\theta}}_i}(\boldsymbol{\pi}|s)$ for subproblem $i$ to select node $i'$ is computed as follows,

$$
P_{i',i} = P_{\tilde{\boldsymbol{\theta}}_i}(\pi_t = i'|\boldsymbol{\pi}_{1:t-1}, s) = \frac{e^{u_{i',i}}}{\sum_{j=1}^{n} e^{u_{j,i}}}. \tag{10}
$$

## B  Scaled symmetric sampling method

The scaled symmetric sampling method is shown in Algorithm 2. The scaled factor $f'_m$ is first estimated (Lines 1 – 3). Then $\lfloor \tilde{N}/M \rfloor$ weight vectors are randomly sampled (Lines 5 – 6). For each of them, $M - 1$ scaled symmetric weight vectors are generated (Lines 7 – 16).

**Algorithm 2** Scaled symmetric sampling method

---

**Input:** meta-model $\boldsymbol{\theta}$, problem size $n$, weight vector distribution $\Lambda$, number of objectives $M$, number of symmetric sampled weight vectors $\tilde{N}$, validation dataset $\mathcal{V}$
1: $\{\boldsymbol{\pi}^k | \mathcal{V}_j\} \sim \textbf{GreedyRollout}(P_{\boldsymbol{\theta}}(\cdot | \mathcal{V}_j)), \quad \forall j \in \{1, \cdots, |\mathcal{V}|\}, \forall k \in \{1, \cdots, n\}$
2: $\boldsymbol{\pi}_j \leftarrow \text{argmax}_k \, g(\boldsymbol{\pi}^k | (\mathcal{V}_j, \mathbf{1}/M))$
3: $f'_m \leftarrow \frac{1}{|\mathcal{V}|} \sum_{j=1}^{|\mathcal{V}|} f_m(\boldsymbol{\pi}_j), \quad \forall m \in \{1, \cdots, M\}$
4: **for** $i = 1$ to $\tilde{N}$ **do**
5:    **if** $i \leq \lfloor \tilde{N}/M \rfloor$ **then**
6:       $\boldsymbol{\lambda}_i \sim \textbf{SampleWeight}(\Lambda)$
7:    **else if** $\lfloor \tilde{N}/M \rfloor < i \leq M \times \lfloor \tilde{N}/M \rfloor$ **then**
8:       **for** $m = 1$ to $M$ **do**
9:          $\boldsymbol{\lambda}'_i \leftarrow \boldsymbol{\lambda}_i$
10:          **if** $m = 1$ **then**
11:             $\lambda_{i,m} \leftarrow \lambda'_{i-\lfloor \tilde{N}/M \rfloor, M} \times f'_M / f'_m$
12:          **else**
13:             $\lambda_{i,m} \leftarrow \lambda'_{i-\lfloor \tilde{N}/M \rfloor, m-1} \times f'_{m-1} / f'_m$
14:          **end if**
15:       **end for**
16:       $\boldsymbol{\lambda}_i \leftarrow \boldsymbol{\lambda}_i / \sum_{m=1}^{M} \lambda_{i,m}$
17:    **else**
18:       $\boldsymbol{\lambda}_i \sim \textbf{SampleWeight}(\Lambda)$
19:    **end if**
20: **end for**
**Output:** $\{\boldsymbol{\lambda}_1, \cdots, \boldsymbol{\lambda}_{\tilde{N}}\}$

---

## C   Hierarchical fine-tuning method

We consider an $L$-level $a$-section hierarchy. The whole weight space is uniformly divided into $N^{(l)}$ subspaces in level $l$. $N^{(l)}$ weight vectors are the centers of these subspaces. In level $l + 1$, the $N^{(l)}$ submodels are fine-tuned to derive $N^{(l+1)} = aN^{(l)}$ submodels with $K^{(l+1)}$ fine-tuning steps, which are the centers of $N^{(l+1)}$ subspaces. The $j$-th submodel in level $l + 1$ is fine-tuned from the $i$-th submodel in level $l$, where the $j$-th weight vector belongs to the $i$-th subspace in level $l$.

The uniform division of the weight space is illustrated as follows. According to the Das and Dennis [44] method, $C_{H+M-1}^{M-1}$ vertices can be generated, where $M$ is the number of objectives and $H$ is a user-defined hyper-parameter. Then, the whole weight space is uniformly divided by $C_{H+M-1}^{M-1}$ vertices. The division with $H^{(l)} = 2^l$ can be seen in Figure 5. There are $2^l$ subspaces for $M = 2$ and $4^l$ subspaces for $M = 3$ in level $l$.

Note that the final $N$ weights are given beforehand. Thus, we set $N^{(L)} = N$, where $N^{(L-1)} = a^{L-1} < N$ and $a^L \geq N$, i.e., $L = 7$ for $M = 2$ and $L = 4$ for $M = 3$. The final $N$ submodels are fine-tuned by $N^{(L-1)}$ submodels.

For the meta-model or a coarse-tuned submodel in level $l$ ($l < L$), its fine-tuning process with a given weight vector and $K$ fine-tuning steps is shown in Algorithm 3.

## D   Details of MOCOPs

### D.1   MOTSP

#### D.1.1   Problem definition

The multi-objective traveling salesman problem (MOTSP) is defined on a complete graph with $n$ nodes and $M$ cost matrices. Node $i$ has $M$ 2D coordinates $\{\boldsymbol{x}_i^1, \ldots, \boldsymbol{x}_i^M\}$, where the $m$-th cost between node $i$ and $j$ is given by the Euclidean distance $c_{ij}^m = \|\boldsymbol{x}_i^m - \boldsymbol{x}_j^m\|_2$. The goal is to find a

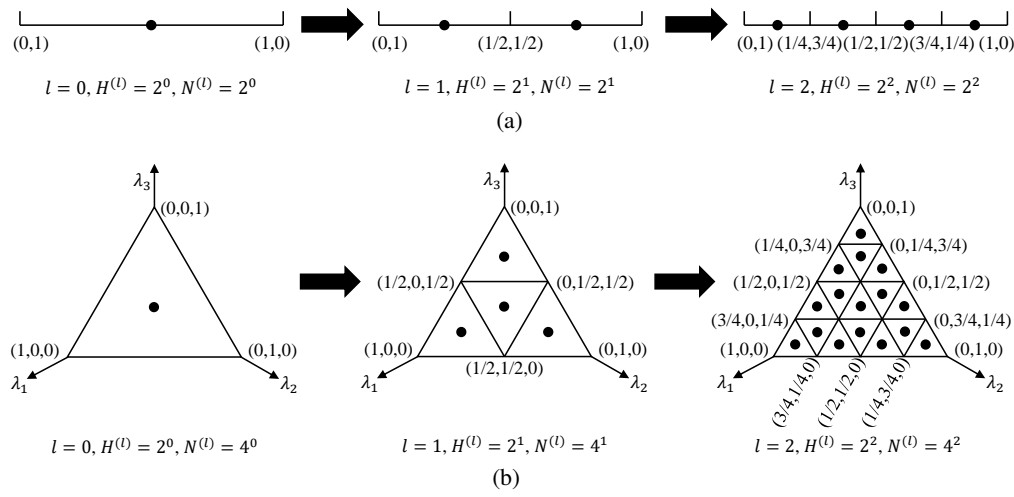

Figure 5: The hierarchy of fine-tuning. (a) $M = 2$. (b) $M = 3$.

---

**Algorithm 3** Fine-tuning process in each level

---

**Input:** the meta-model or a coarse-tuned submodel $\boldsymbol{\theta}$, a given weight vector $\boldsymbol{\lambda}$, fine-tuning steps $K$, batch size $B$, problem size $n$
1: **for** $k = 1$ to $K$ **do**
2:     $s_j \sim \textbf{SampleInstance}(\mathcal{S}), \quad \forall j \in \{1, \dots, B\}$
3:     $\{\boldsymbol{\pi}^k | s_j, \boldsymbol{\lambda}\} \sim \textbf{SampleRollout}(P_{\tilde{\boldsymbol{\theta}}_i}(\cdot | s_j)), \quad \forall j \in \{1, \dots, B\}, \forall k \in \{1, \dots, n\}$
4:     $b_j \leftarrow \frac{1}{n} \sum_{k=1}^{n} g(\boldsymbol{\pi}^k | s_j, \boldsymbol{\lambda})$
5:     $\nabla \mathcal{L}(\tilde{\boldsymbol{\theta}}) \leftarrow \frac{1}{Bn} \sum_{j=1}^{B} \sum_{k=1}^{n} [(g(\boldsymbol{\pi}^k | s_j, \boldsymbol{\lambda}) - b_j) \nabla \log P_{\boldsymbol{\theta}}(\boldsymbol{\pi}^k | s_j)]$
6:     $\boldsymbol{\theta} \leftarrow \textbf{Adam}(\boldsymbol{\theta}, \nabla \mathcal{L}(\boldsymbol{\theta}))$
7: **end for**
**Output:** The fine-tuned submodel $\boldsymbol{\theta}$

---

permutation $\boldsymbol{\pi}$ to minimize all the $M$ costs simultaneously, as follows,

$$\min \ \boldsymbol{f}(\boldsymbol{\pi}) = (f_1(\boldsymbol{\pi}), f_2(\boldsymbol{\pi}), \dots, f_M(\boldsymbol{\pi})), \tag{11}$$

$$\text{where } f_m(\boldsymbol{\pi}) = c_{\pi_n, \pi_1}^m + \sum_{j=1}^{n-1} c_{\pi_j, \pi_{j+1}}^m. \tag{12}$$

For the $M$-objective TSP type 2, each node has $(M - 1)$ 2D coordinates and one number interpreted as its altitude. The single-objective counterpart, TSP, is a well-known NP-hard problem. It appears that MOTSP is even harder. Thus, its approximate Pareto optimal solutions are commonly pursued.

### D.1.2 Instance

For the $M$-objective TSP type 1, each node has $M$ 2D coordinates. The random instances with $n$ nodes are sampled from uniform distribution on $[0, 1]^{2M}$. For the $M$-objective TSP type 2, each node has $(M - 1)$ 2D coordinates and one number. The random instances with $n$ nodes are sampled from uniform distribution on $[0, 1]^{2M-1}$.

### D.1.3 Model details

The input dimension of the $M$-objective TSP type 1 is $2M$ for the encoder. The input dimension of the $M$-objective TSP type 2 is $2M - 1$ for the encoder.

For MOTSP, POMO [26] uses $n$ context embedding $\boldsymbol{h}_c^1, \ldots, \boldsymbol{h}_c^n$ to calculate the probability of node selection in the decoder. At the decoding step $t$, $\boldsymbol{h}_c^k$ is defined as follows,

$$\boldsymbol{h}_c^k = \begin{cases} [\bar{\boldsymbol{h}}^k; \boldsymbol{h}_{\boldsymbol{\pi}_{t-1}}^k; \boldsymbol{h}_{\boldsymbol{\pi}_1}^k], & t > 1 \\ \text{none}, & t = 1 \end{cases} \tag{13}$$

where $[;]$ is the concatenation and the graph embedding $\bar{\boldsymbol{h}}^k = \sum_{i=1}^n \boldsymbol{h}_i^k$. For $t = 1$, the first node is not selected by the decoder. Instead, it is defined as $\boldsymbol{h}_{\boldsymbol{\pi}_1}^k = \boldsymbol{h}_k$. In the decoding procedure, the nodes already visited need to be masked.

## D.2 MOCVRP

### D.2.1 Problem definition

The capacitated vehicle routing problem (CVRP), a classical extension of TSP, contains $n$ customer nodes and a depot node. Each node has a 2D coordinate. In addition, customer $i$ has a demand $d_i$ to be satisfied. A fleet of homogeneous vehicles with identical capacity $D$ is initially placed at the depot. Vehicles must serve all the customers and finally return to the depot. The capacity constraints must be satisfied, i.e., the remaining capacity of vehicles for serving customer $i$ must be no less than $d_i$.

For the multi-objective capacitated vehicle routing problem (MOCVRP), we consider two conflicting objectives, i.e., the total traveling distance and the traveling distance of the longest route (makespan).

### D.2.2 Instance

For MOCVRP, the coordinates of the depot and customers are sampled from uniform distribution on $[0, 1]^2$. Following the previous work [26, 12], the demand $d_i$ is randomly chosen from $\{1, \ldots, 9\}$ and the capacity is set to $D = 30/40/50$ for $n = 20/50/100$. Without loss of generality, the demand and capacity are normalized as $\hat{d}_i = d_i/D$ and $\hat{D} = D/D = 1$, respectively.

### D.2.3 Model details

The inputs of MOCVRP are a 2D vector of the depot and $n$ 3D vectors of the customers for the encoder. Their embeddings with both 128 dimensions are obtained by two linear projections with two separate parameter matrices.

For MOCVRP, the context embedding in the decoder is defined as the concatenation of the graph embedding, the embedding of the last node, and the remaining capacity. Since the first node of CVRP must be the depot node, POMO assigns the second node, i.e., the first customer node, to produce $n$ various solutions. In the decoding procedure, the nodes already visited and the nodes with a larger demand than the remaining capacity need to be masked.

## D.3 MOKP

### D.3.1 Problem definition

Knapsack problem (KP) is also a representative combinatorial optimization problem. The multi-objective 0-1 knapsack problem (MOKP) with $M$ objectives and $n$ items is defined as follows,

$$\max \boldsymbol{f}(\boldsymbol{x}) = (f_1(\boldsymbol{x}), f_2(\boldsymbol{x}), \ldots, f_M(\boldsymbol{x})), \tag{14}$$

$$\text{where } f_m(\boldsymbol{x}) = \sum_{i=1}^n v_i^m x_i, \tag{15}$$

$$\text{subject to } \sum_{i=1}^n w_i x_i \leq W, \tag{16}$$

$$x_i \in \{0, 1\}, \tag{17}$$

where item $i$ has a weight $w_i$ and $M$ different values $\{v_i^1, v_i^2, \ldots, v_i^M\}$. $W$ is the weight capacity of the knapsack. The goal is to find a solution $\boldsymbol{x}$ to maximize all the $M$ total values simultaneously. The single-objective KP is also an NP-hard problem, so MOKP is even harder.

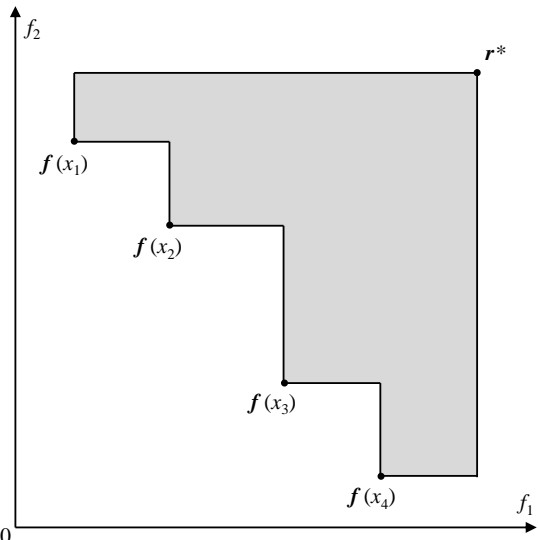

Table 3: Hypervolume illustration.

Table 4: Reference points and ideal points.

| Problem | Size | $r^*$ | $z^*$ |
|---|---|---|---|
| Bi-TSP-1 | 20 | (20, 20) | (0, 0) |
| | 50 | (35, 35) | (0, 0) |
| | 100 | (65, 65) | (0, 0) |
| | 150 | (85, 85) | (0, 0) |
| | 200 | (115, 115) | (0, 0) |
| Bi-KP | 50 | (5, 5) | (30, 30) |
| | 100 | (20, 20) | (50, 50) |
| | 200 | (30, 30) | (75, 75) |
| Tri-TSP-1 | 20 | (20, 20, 20) | (0, 0) |
| | 50 | (35, 35, 35) | (0, 0) |
| | 100 | (65, 65, 65) | (0, 0) |
| Bi-CVRP | 20 | (30, 4) | (0, 0) |
| | 50 | (45, 4) | (0, 0) |
| | 100 | (80, 4) | (0, 0) |
| Bi-TSP-2 | 20 | (20, 12) | (0, 0) |
| | 50 | (35, 25) | (0, 0) |
| | 100 | (65, 45) | (0, 0) |
| Tri-TSP-2 | 20 | (20, 20, 12) | (0, 0) |
| | 50 | (35, 35, 25) | (0, 0) |
| | 100 | (65, 65, 45) | (0, 0) |

#### D.3.2 Instance

As in the previous work [26, 12], the values and weight for each item of MOKP are all sampled from uniform distribution on $[0, 1]$. The knapsack capacity is set to $W = 12.5/25/25$ for $n = 50/100/200$.

#### D.3.3 Model details

For the $M$-objective MOKP, as each item has $M$ values and one weight, the input dimension is $M + 1$ for the encoder. The policy network of MOKP is as the same as that of MOTSP. In the decoding procedure, the context embedding is defined as the concatenation of the graph embedding and the remaining capacity. The items already selected and the items with a larger weight than the remaining capacity need to be masked.

### E Hypervolume indicator

The hypervolume (HV) indicator is widely used to evaluate the performance of the methods for MOCOPs, since HV can comprehensively measure the convergence and diversity of $\mathcal{PF}$ without the ground truth Pareto front [51]. For a given $M$-objective $\mathcal{PF}$ and a reference point $r^*$, $\text{HV}(\mathcal{PF}, r^*)$ is defined as follows,

$$\text{HV}(\mathcal{PF}, r^*) = \mu(S), \tag{18}$$

$$S = \{r \in \mathcal{R}^M | \exists r \in \mathcal{PF} \text{ such that } y \prec r \prec r^*\}, \tag{19}$$

where $r_i^* > \max\{f_i(x)|f(x) \in \mathcal{PF}\}$ (or $r_i^* < \min\{f_i(x)|f(x) \in \mathcal{PF}\}$ for the maximization problem), $\forall i \in \{1, \ldots, M\}$, and $\mu$ is the Lebesgue measure. An example with $M = 2$ is presented in Figure 3, where $\mathcal{PF} = \{f(x_1), f(x_2), f(x_3), f(x_4)\}$. $\text{HV}(\mathcal{PF}, r^*)$ is equal to the size of the grey area in this example.

Since HV value would considerably vary with the objective domain, we record the normalized HV ratio $\text{HV}'(\mathcal{PF}, r^*) = \text{HV}(\mathcal{PF}, r^*) / \prod_{i=1}^{M} |r_i^* - z_i^*|$, where $z^*$ is an ideal point satisfying $z_i^* < \min\{f_i(x)|f(x) \in \mathcal{PF}\}$ (or $z_i^* > \max\{f_i(x)|f(x) \in \mathcal{PF}\}$ for the maximization problem), $\forall i \in \{1, \ldots, M\}$. For a problem, all methods share the same $r^*$ and $z^*$, as shown in Table 4.

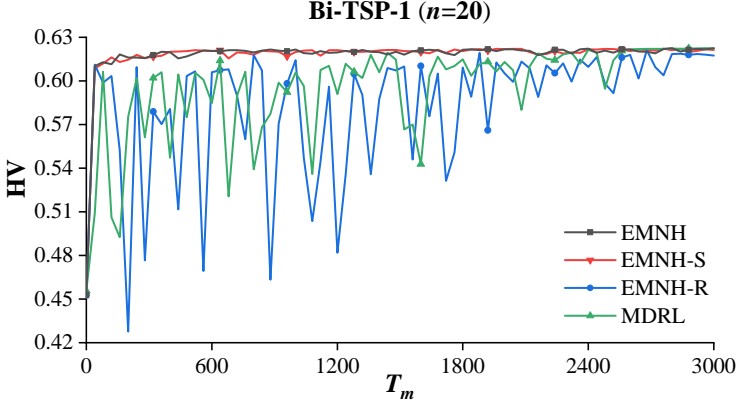

Figure 6: Training Stability on MOCOPs with balanced objective domains.

# F More details of learning efficiency

## F.1 Training efficiency

EMNH uses almost the equal number of training instances to MDRL [13] and PMOCO [12]. Attributed to the accelerated training method, EMNH only consumes approximately $1/\tilde{N}$ training time of MDRL. For EMNH, we set $T_m = 3000$, $T_u = 100$, and $B = 64$ to train the meta-model, i.e., overall $1.92 \times 10^7$ training instances. MDRL adopts the same training hyper-parameters as EMNH, i.e., the same amount of training instances. According to the settings of PMOCO, it trains the model by 200 epochs, with $1 \times 10^5$ instances at each epoch, i.e., overall $2 \times 10^7$ training instances, a little more than that of EMNH. The total training time of PMOCO is close to that of EMNH.

As a multi-model method, DRL-MOA [8] needs to train multiple networks with more training instances to deal with multiple subproblems. DRL-MOA first trains a submodel for the first weight vector by 200 epochs with $1 \times 10^5$ instances at each epoch, and then transfers its parameter by 5 epochs to derive another submodel for its neighbor subproblem. With a sequential parameter-transfer process, a set of submodels are obtained for various pre-given weight vectors. For the bi/tri-objective problems, the number of subproblems $N$ is set to 101/105. Hence, DRL-MOA needs to train 101/105 submodels with total 700/720 training epochs.

## F.2 Training stability

Figure 6 shows the results on Bi-TSP-1 with the balanced objective domains. EMNH-S and EMNH-R denote EMNH with the symmetric sampling method and the random sampling method, respectively. HV is computed by fine-tuning the meta-model with $K = 1$ step at each level. The results show that EMNH achieves the stablest training process. The performance of EMNH and EMNH-S are close, while the training processes of other two methods with random sampling are unstable. However, for Bi-CVRP with the imbalanced objective domains, EMNH, which adjusts the sampled weight vectors by the objective domains, outperforms other methods, as shown in Figure 4(b).

Table 5 reports the final HV indicators of the trained model, which also exhibits higher solution quality of EMNH. The best result and its statistically indifferent results using a Wilcoxon rank-sum test at 1% level are highlighted as **bold**. The second-best result and its statistically indifferent results are highlighted as underline. The method with "-Aug" represents the inference results using instance augmentation (see Appendix G) of POMO.

## F.3 Fine-tuning efficiency

To study the fine-tuning efficiency, we compare the hierarchical fine-tuning method of EMNH with the vanilla fine-tuning method of MDRL. For $K$ steps at each level with $M = 2$ in EMNH, the total fine-tuning step is $\sum_{l=1}^{L-1} 2^l K + NK = 227K$ with $N = 101$, where $L$ satisfies $2^{L-1} < N$ and $2^L \geq N$, i.e., $L = 7$. For MDRL, each submodel is fine-tuned directly from the meta-model with $\tilde{K}$ steps,

Table 5: Solution quality of the scaled symmetric sampling method.

| | Method | $n$=20 | | $n$=50 | |
|---|---|---|---|---|---|
| | | HV↑ | Gap↓ | HV↑ | Gap↓ |
| Bi-TSP-1 | MDRL | **0.6271** | **0.00%** | 0.6374 | 0.53% |
| | EMNH-R | 0.6269 | 0.04% | 0.6339 | 1.07% |
| | EMNH-S | 0.6270 | 0.02% | 0.6362 | 0.72% |
| | EMNH | **0.6271** | **0.00%** | 0.6362 | 0.72% |
| | MDRL-Aug | **0.6271** | **0.00%** | **0.6408** | **0.00%** |
| | EMNH-R-Aug | 0.6269 | 0.04% | 0.6391 | 0.26% |
| | EMNH-S-Aug | 0.6270 | 0.02% | **0.6408** | **0.00%** |
| | EMNH-Aug | **0.6271** | **0.00%** | **0.6408** | **0.00%** |
| Bi-CVRP | MDRL | 0.4291 | 0.26% | 0.4082 | 0.58% |
| | EMNH-R | 0.3774 | 12.26% | 0.3743 | 8.83% |
| | EMNH-S | 0.4158 | 3.34% | 0.3825 | 6.84% |
| | EMNH | 0.4299 | 0.07% | 0.4098 | 0.19% |
| | MDRL-Aug | 0.4294 | 0.19% | 0.4092 | 0.34% |
| | EMNH-R-Aug | 0.3849 | 10.53% | 0.3786 | 7.78% |
| | EMNH-S-Aug | 0.4179 | 2.87% | 0.3870 | 5.76% |
| | EMNH-Aug | **0.4302** | **0.00%** | **0.4106** | **0.00%** |

so the total fine-tuning step is $N\tilde{K} = 101\tilde{K}$. Thus, for $K = 1/5/10/15/20$, $\tilde{K} = 2/11/22/34/45$ ensures that the total fine-tuning step of MDRL is approximately equal to that of EMNH. For $M = 3$ with $N = 105$, $K$ is set to 25 for EMNH, while $\tilde{K}$ is set to 45 for MDRL to make the equal total of fine-tuning steps.

Figure 7 shows that, on various MOCOPs, EMNH attains better solution quality than MDRL with equal total fine-tuning steps, especially a larger gap with a few fine-tuning steps, which means that EMNH has higher fine-tuning efficiency. Furthermore, PMOCO is equipped with the hierarchical fine-tuning method for a fair comparison. However, it could hardly get improved, which indicates that EMNH has more potential to derive performant submodels to deal with specific tasks. Besides, in most cases, EMNH outperforms PMOCO only with a few fine-tuning steps ($K = 5$).

## G    Instance augmentation

Instance augmentation exploits multiple efficient transformations for the original instance that share the same optimal solution. Then, all transformed problems are solved and the best solution among them are finally selected. According to POMO [26], a 2D coordinate $(x, y)$ has eight different transformations, $\{(x, y), (y, x), (x, 1-y), (y, 1-x), (1-x, y), (1-y, x), (1-x, 1-y), (1-y, 1-x)\}$.

For the $M$-objective TSP type 1, each node has $M$ 2D coordinates, so it has $8^M$ different transformations. For the $M$-objective TSP type 2, each node has $(M-1)$ 2D coordinates and a 1D coordinate. The 1D coordinate $x$ has two different transformations, $\{x, 1-x\}$. Thus, the $M$-objective TSP type 2 has $2 \times 8^{M-1}$ different transformations. For MOCVRP, each node has a 2D coordinate, so it has $8$ different transformations. MOKP has no transformation.

## H    More results on MOCOPs with imbalanced objective domains

### H.1    Scaled weight sampling method for training

For the problems with imbalanced objective domains, including Bi-CVRP, Bi-TSP-2, and Tri-TSP-2, we further study PMOCO with a scaled sampling method in the training phase, denoted as PMOCO-S. Specifically, in each sampling for PMOCO, a sampled weight vector $\boldsymbol{\lambda}$ is sacled by $\boldsymbol{f}'$ as $\lambda_m^s = \lambda_m/f_m'$, where $f_m'$ is dynamically estimated by the model on a validation dataset associated with $\boldsymbol{\lambda} = (1/M, \ldots, 1/M)$ during training. Table 6 shows that PMOCO-S could certainly improve the performance of PMOCO for Bi-CVRP and Tri-TSP-2, but it is still inferior to EMNH.

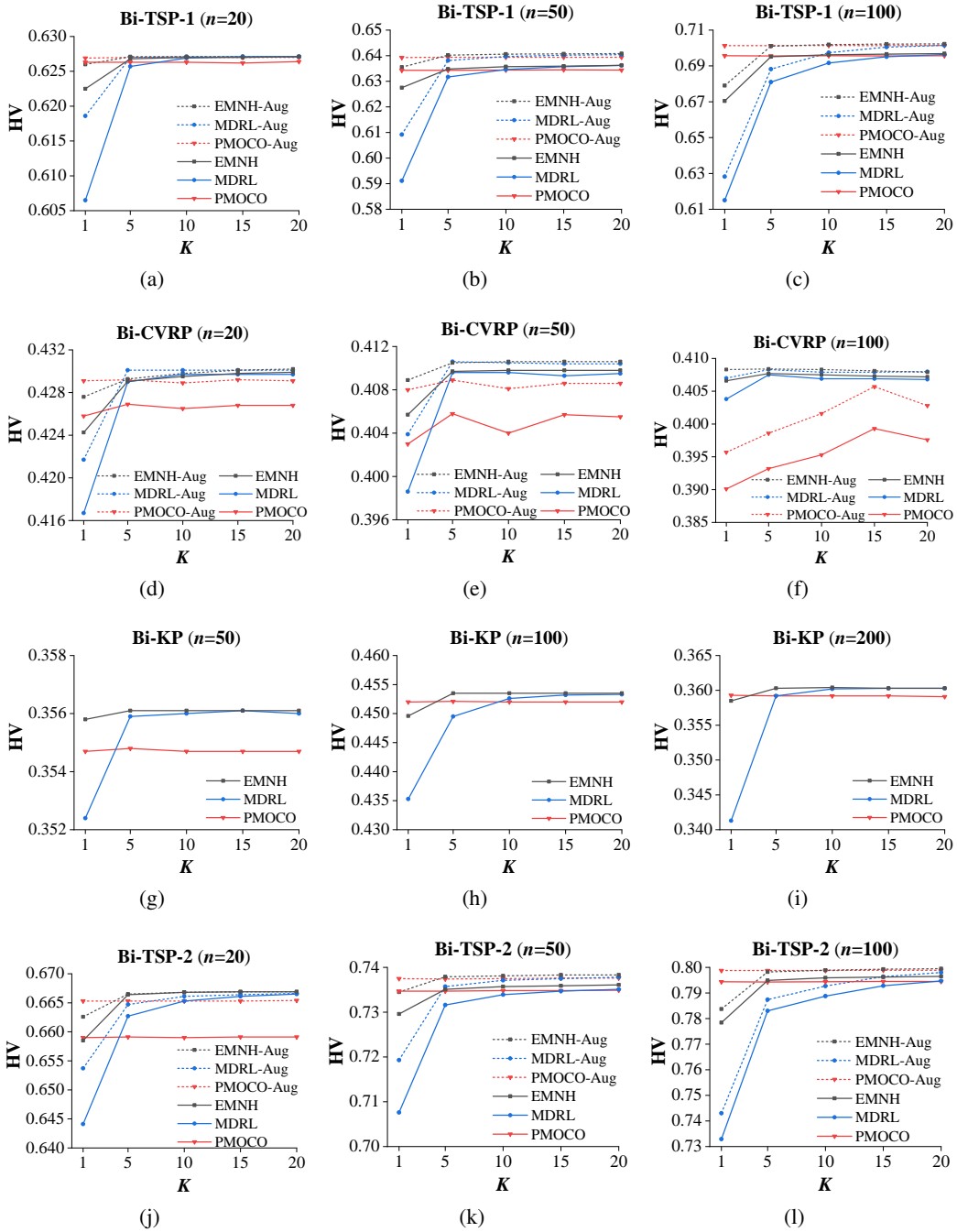

Figure 7: Fine-tuning efficiency on various MOCOPs.

## H.2 Scaled weight assignment method for inference

Similar to other decomposition-based methods like PMOCO and MDRL, EMNH also faces the challenge of non-uniformly solution distributions. However, this issue can be mitigated by employing appropriate weight assignment methods during inference, as EMNH offers the flexibility to handle arbitrary weight vectors. Specifically, when the approximate scales of different objectives are known, we can normalize them to [0,1] to achieve a more uniform Pareto front. Alternatively, we can adjust the weight assignment to generate a more uniform Pareto front.

Table 6: PMOCO with the scaled sampling method.

| | Method | n=20 HV↑ | n=20 Gap↓ | n=50 HV↑ | n=50 Gap↓ | n=100 HV↑ | n=100 Gap↓ |
|---|---|---|---|---|---|---|---|
| Bi-CVRP | PMOCO | 0.4267 | 0.81% | 0.4036 | 1.70% | 0.3913 | 4.07% |
| | PMOCO-S | 0.4274 | 0.65% | 0.4057 | 1.19% | 0.4042 | 0.91% |
| | EMNH | 0.4299 | 0.07% | 0.4098 | 0.19% | 0.4072 | 0.17% |
| | PMOCO-Aug | 0.4294 | 0.19% | 0.4080 | 0.63% | 0.3969 | 2.70% |
| | PMOCO-S-Aug | 0.4295 | 0.16% | 0.4090 | 0.39% | 0.4063 | 0.39% |
| | EMNH-Aug | **0.4302** | **0.00%** | **0.4106** | **0.00%** | **0.4079** | **0.00%** |
| Bi-TSP-2 | PMOCO | 0.6590 | 1.18% | 0.7347 | 0.51% | 0.7944 | 0.65% |
| | PMOCO-S | 0.6520 | 2.23% | 0.7333 | 0.70% | 0.7927 | 0.86% |
| | EMNH | **0.6669** | **0.00%** | 0.7361 | 0.32% | 0.7965 | 0.39% |
| | PMOCO-Aug | 0.6653 | 0.24% | 0.7375 | 0.14% | 0.7988 | 0.10% |
| | PMOCO-S-Aug | 0.6624 | 0.67% | 0.7366 | 0.26% | 0.7974 | 0.28% |
| | EMNH-Aug | **0.6669** | **0.00%** | **0.7385** | **0.00%** | **0.7996** | **0.00%** |
| Tri-TSP-2 | PMOCO | 0.5020 | 0.30% | 0.5176 | 1.95% | 0.5777 | 2.02% |
| | PMOCO-S | 0.4917 | 2.34% | 0.5175 | 1.97% | 0.5778 | 2.00% |
| | EMNH | 0.5022 | 0.26% | 0.5205 | 1.40% | 0.5813 | 1.41% |
| | PMOCO-Aug | **0.5035** | **0.00%** | 0.5258 | 0.40% | 0.5862 | 0.58% |
| | PMOCO-S-Aug | 0.5029 | 0.12% | 0.5259 | 0.38% | 0.5863 | 0.56% |
| | EMNH-Aug | **0.5035** | **0.00%** | **0.5279** | **0.00%** | **0.5896** | **0.00%** |

A scaled weight assignment (SWA) method can be directly applied to alleviate this issue. Specifically, each uniform weight vector $\boldsymbol{\lambda}$ is scaled by $\boldsymbol{f}'$ as $\lambda_m^s = \lambda_m/f'_m$ and normalized to $[0,1]^M$. Here, $\boldsymbol{f}'$ is estimated using a validation dataset associated with $\boldsymbol{\lambda} = (1/M, \ldots, 1/M)$. The advantage of this SWA method is that it does not require prior problem information.

The results on Tri-TSP instances with asymmetric Pareto fronts are presented in Figure 8. For these instances, the coordinates for the three objectives are randomly sampled from $[0,1]^2$, $[0,0.5]^2$, $[0,0.1]^2$, respectively. The results demonstrate that EMNH-SWA effectively produces a more uniform Pareto front. Compared to a scaling weight method with prior knowledge used in PMOCO [12], where uniform weight vectors are element-wise multiplied by (1,2,10) and then normalized back to $[0,1]^3$, EMNH-SWA achieves desirable performance.

# I  Detailed results of generalization capability

We test the generalization capability of our EMNH on larger-scale random instances ($n$=150/200) and three commonly used MOTSP benchmark instances (KroAB100/150/200) in TSPLIB [52], as shown in Tables 7 and 8, respectively. EMNH, which is trained and fine-tuned both on the instances with $n$=100, has a superior generalization capability compared with the state-of-the-art MOEA and other neural heuristics for larger problem sizes.

# J  Hyper-parameter study

## J.1  Number of sampled weight vectors

Figure 9 shows the results for various $\tilde{N}$. For bi-objective problems, $\tilde{N} = 1$ leads to a significantly unstable training process, since the sample number is quite small and it has no symmetric sample. $\tilde{N} = 3$ also causes a slightly unstable training process. $\tilde{N} = kM$ with $k \in \{1, 2, \ldots\}$ can effectively stabilize the training process. In summary, $\tilde{N} = M$ is a more favorable setting.

## J.2  Scalarization method

EMNH is generic for solving MOCOPs based on decomposition [17], which can employ various scalarization methods, including weighted sum (WS) and Tchebycheff (TCH).

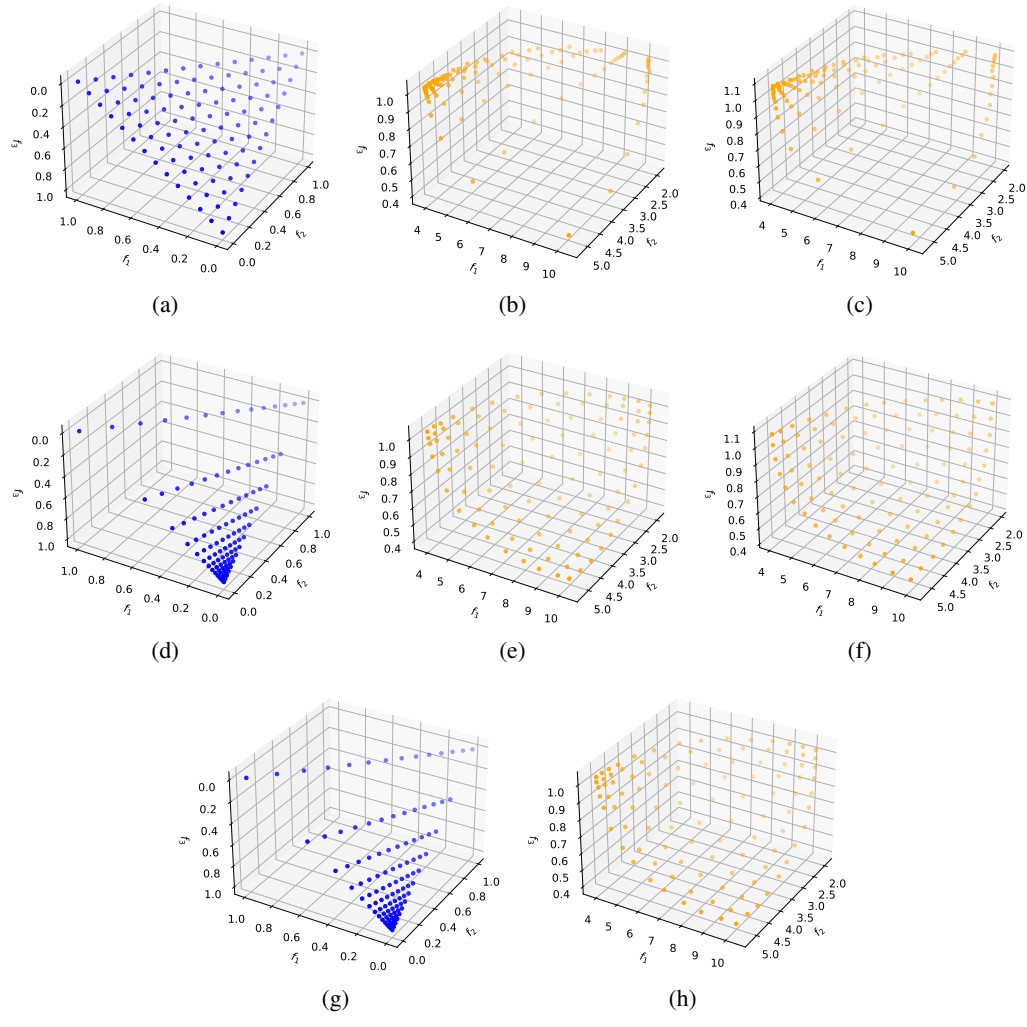

Figure 8: Solutions generated by using 105 uniform/non-uniform distributed weights on Tri-TSP-1 ($n$=20) with asymmetric Pareto front. (a) Uniform weights. (b) EMNH with uniform weights. (c) PMOCO with uniform weights. (d) A priori non-uniform weights. (e) EMNH with a priori non-uniform weights. (f) PMOCO with a priori non-uniform weights. (g) Weights generated by the scaled weight assignment (SWA) method. (h) EMNH with SWA.

Table 7: Results of generalization capability on 200 random instances .

| Method | Bi-TSP-1 ($n$=150) | | | Bi-TSP-1 ($n$=200) | | |
|---|---|---|---|---|---|---|
| | HV↑ | Gap↓ | Time | HV↑ | Gap↓ | Time |
| WS-LKH | **0.7149** | **-2.38%** | 13h | **0.7490** | **-2.50%** | 22h |
| PPLS/D-C | 0.6784 | 3.25% | 21h | 0.7106 | 3.11% | 32h |
| DRL-MOA | 0.6901 | 1.17% | 45s | 0.7219 | 1.20% | 87s |
| PMOCO | 0.6910 | 1.05% | 45s | 0.7231 | 1.04% | 87s |
| MDRL | 0.6922 | 0.87% | 40s | 0.7251 | 0.77% | 84s |
| EMNH | 0.6930 | 0.76% | 40s | 0.7260 | 0.64% | 83s |
| PMOCO-Aug | 0.6967 | 0.23% | 47m | 0.7283 | 0.33% | 1.6h |
| MDRL-Aug | 0.6976 | 0.10% | 47m | 0.7299 | 0.11% | 1.6h |
| EMNH-Aug | 0.6983 | 0.00% | 47m | 0.7307 | 0.00% | 1.6h |

Table 8: Results of generalization capability on benchmark instances.

| Method | KroAB100 HV↑ | KroAB100 Gap↓ | KroAB100 Time | KroAB150 HV↑ | KroAB150 Gap↓ | KroAB150 Time | KroAB200 HV↑ | KroAB200 Gap↓ | KroAB200 Time |
|---|---|---|---|---|---|---|---|---|---|
| WS-LKH | **0.7022** | **-0.92%** | 2.3m | **0.7017** | **-1.81%** | 4.0m | **0.7430** | **-2.20%** | 5.6m |
| PPLS/D-C | 0.6785 | 2.77% | 31m | 0.6659 | 3.84% | 1.1h | 0.7100 | 2.69% | 3.1h |
| DRL-MOA | 0.6903 | 0.79% | 10s | 0.6794 | 1.42% | 18s | 0.7185 | 1.17% | 23s |
| PMOCO | 0.6878 | 1.15% | 9s | 0.6819 | 1.06% | 17s | 0.7193 | 1.06% | 23s |
| MDRL | 0.6881 | 1.11% | 10s | 0.6831 | 0.89% | 17s | 0.7209 | 0.84% | 23s |
| EMNH | 0.6900 | 0.83% | 9s | 0.6832 | 0.87% | 16s | 0.7217 | 0.73% | 23s |
| PMOCO-Aug | 0.6937 | 0.30% | 12s | 0.6886 | 0.09% | 19s | 0.7251 | 0.26% | 27s |
| MDRL-Aug | 0.6950 | 0.11% | 13s | 0.6890 | 0.03% | 19s | 0.7261 | 0.12% | 28s |
| EMNH-Aug | 0.6958 | 0.00% | 12s | 0.6892 | 0.00% | 18s | 0.7270 | 0.00% | 27s |

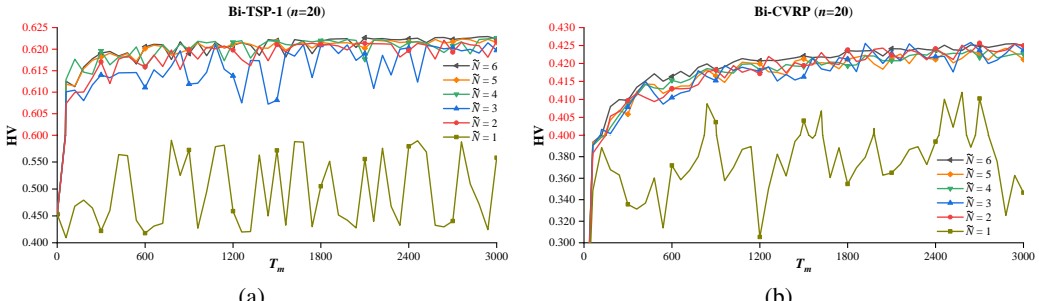

Figure 9: Effects of the number of sampled weight vectors.

Table 9: Effects of the scalarization method.

| | Method | n=20 HV↑ | n=20 Gap↓ | n=50 HV↑ | n=50 Gap↓ | n=100 HV↑ | n=100 Gap↓ |
|---|---|---|---|---|---|---|---|
| Bi-TSP-1 | EMNH-TCH | 0.6271 | 0.00% | 0.6331 | 1.20% | 0.6927 | 1.37% |
| | EMNH-WS | 0.6271 | 0.00% | 0.6364 | 0.69% | 0.6969 | 0.77% |
| | EMNH-TCH-Aug | **0.6294** | **-0.37%** | 0.6401 | 0.11% | 0.6995 | 0.40% |
| | EMNH-WS-Aug | 0.6271 | 0.00% | **0.6408** | **0.00%** | **0.7023** | **0.00%** |
| Tri-TSP-1 | EMNH-TCH | 0.4665 | 1.00% | 0.4175 | 5.50% | 0.4681 | 5.87% |
| | EMNH-WS | 0.4698 | 0.30% | 0.4324 | 2.13% | 0.4866 | 2.15% |
| | EMNH-TCH-Aug | 0.4710 | 0.04% | 0.4295 | 2.78% | 0.4814 | 3.20% |
| | EMNH-WS-Aug | **0.4712** | **0.00%** | **0.4418** | **0.00%** | **0.4973** | **0.00%** |

WS is the simplest method and is effective for the convex $\mathcal{PF}$. It considers the linear combination of $M$ objectives, as follows,

$$\min_{x \in \mathcal{X}} g_{\mathrm{ws}}(x|\boldsymbol{\lambda}) = \sum_{m=1}^{M} \lambda_m f_m(x). \tag{20}$$

In theory, TCH can tackle the concave $\mathcal{PF}$, but it would lead to a more complex objective function. It is defined as follows,

$$\min_{x \in \mathcal{X}} g_{\mathrm{tch}}(x|\boldsymbol{\lambda}) = \max_{1 \leq m \leq M} \{\lambda_m |f_m(x) - z_m^*|\}, \tag{21}$$

where $z_m^* < \min_{x \in \mathcal{X}} f_m(x)$ is an ideal value of $f_m(x)$.

As shown in Table 9, the results show that WS is a simple yet effective scalarization method for the studied problems. In principle, EMNH can freely use any existing scalarizing function. Different scalarization methods have their own strengths and drawbacks, but the study with respect to scalarization methods is beyond the scope of this paper, which we will investigate in the future.

Table 10: Results of lightweight fine-tuning methods.

| | Method | n=20 HV | n=20 Gap | n=100 HV | n=100 Gap |
|---|---|---|---|---|---|
| Bi-CVRP | PMOCO-Aug | 0.4294 | 0.19% | 0.3966 | 2.77% |
| | MDRL-Aug | 0.4292 | 0.23% | 0.4072 | 0.17% |
| | EMDRL-Aug | 0.4302 | 0.00% | 0.4079 | 0.00% |
| | EMDRL-FD-Aug | 0.4299 | 0.07% | 0.4082 | -0.07% |
| | EMDRL-FH-Aug | 0.4298 | 0.09% | 0.4082 | -0.07% |
| Tri-TSP-1 | PMOCO-Aug | 0.4712 | 0.00% | 0.4956 | 0.34% |
| | MDRL-Aug | 0.4712 | 0.00% | 0.4958 | 0.30% |
| | EMDRL-Aug | 0.4712 | 0.00% | 0.4973 | 0.00% |
| | EMDRL-FD-Aug | 0.4710 | 0.04% | 0.4925 | 0.97% |
| | EMDRL-FH-Aug | 0.4707 | 0.11% | 0.4906 | 1.35% |

Table 11: Parameter numbers of various parts of models.

| | Bi-CVRP Model Whole Model | Decoder | Head | Tri-TSP-1 Model Whole Model | Decoder | Head |
|---|---|---|---|---|---|---|
| #(Parameters) | 1287K | 98K | 16K | 1303K | 115K | 16K |

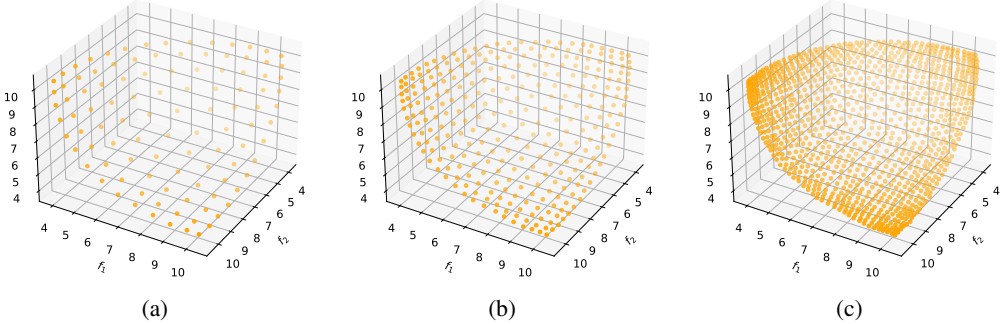

Figure 10: Solutions of Tri-TSP-1 ($n$=20) generated with various numbers of weight vectors. (a) 105 weights. (b) 300 weights. (c) 1035 weights.

## K    Trade-off between lightweight fine-tuning and performance

For a given weight vector, EMNH fine-tunes the meta-model to derive a submodel to solve the corresponding subproblem. We study another two (relatively) lightweight fine-tuning methods, including only updating the head parameter (denoted as EMNH-FH) according to feature reuse [15] and only updating the decoder parameter (denoted as EMNH-FD) like PMOCO [12]. These two methods even allow us to only fine-tune and store parts of the original submodels, i.e., $N$ heads or $N$ decoders, thereby being more computationally efficient. Meanwhile, such benefit may bring about performance sacrifices in some cases. We report the results in Table 10 and the parameter numbers of various parts of the model in Table 11. The lightweight fine-tuning has slightly inferior performance compared with the original EMNH in most cases except on Bi-CVRP ($n$=100). Generally, the more lightweight of the fine-tuning, the more performance deterioration (i.e.,EMNH-FH v.s. EMNH-FD as displayed in the table below, where FH is more light than FD). However, these lightweight fine-tuning methods can be used as alternatives when the computational and memory resources are limited.

Moreover, same as EMNH, EMNH-FH can also generate much more dense Pareto solutions to improve the performance via increasing weight vectors and corresponding fine-tuned heads. We have plotted the generated Pareto fronts with 105, 300 and 1035 weight vectors on Tri-TSP-1 which verified the above point, as shown in Figure 10.