# OpenReview forum: "Efficient Meta Neural Heuristic for Multi-Objective Combinatorial Optimization"
_NeurIPS.cc/2023/Conference — NeurIPS 2023 poster_

### Official Review · Reviewer_EY2k · 2023-07-05

**Soundness:** 3 good
**Presentation:** 3 good
**Contribution:** 2 fair
**Rating:** 6
**Confidence:** 1

**Summary:**

This article presents an efficient neural heuristic method based on meta-learning, referred to as EMNH (Efficient Meta-learning Neural Heuristic), for solving Multi-Objective Combinatorial Optimization Problems (MOCOPs). The authors employ a shared multi-task model to expedite the meta-learning process and introduce a weight vector-based scaling symmetric sampling approach to stabilize this process. Furthermore, they propose a layered fine-tuning method to effectively address all sub-problems. Experimental results demonstrate that EMNH outperforms existing neural heuristic methods on three classical MOCOPs and can also compete with traditional heuristic methods.

**Strengths:**

1. The article addresses the practical and challenging issue of MOCOPs, presenting a novel neural heuristic approach that combines ideas from meta-learning, multi-task learning, and layered fine-tuning.
2. The proposed method's advantages are theoretically analyzed, including accelerated and stabilized meta-learning processes, along with reduced fine-tuning steps.
3. The article conducts comprehensive experiments on three different types of MOCOPs, comparing against various benchmark methods. It showcases the proposed method's advantages in terms of solution quality, learning efficiency, and generalization ability.

**Weaknesses:**

1. The article inadequately discusses the limitations and shortcomings of the proposed method, such as its applicability to more complex or higher-dimensional MOCOPs, sensitivity to weight vector selection, and risks of overfitting or underfitting.
2. The article lacks sufficient details to explain the model architecture, hyperparameter settings, experimental setup, etc. This might affect the replicability and credibility of the results.

**Questions:**

1. Did the authors consider using other types of neural network structures or meta-learning algorithms to implement the proposed method? If so, how do they compare to POMO and Reptile?
2. Did the authors attempt to test the proposed method on other types or scales of MOCOPs? If yes, what were the results? If not, what were the reasons?

**Limitations:**

1. The proposed method might require substantial training data and computational resources to achieve ideal results, potentially limiting its feasibility and scalability in practical applications.
2. The proposed method might struggle with MOCOPs featuring nonlinear or non-convex objective functions, dynamic or stochastic constraints, multi-modal or multi-peak distributions, and other complex features.
3. The proposed method might not guarantee the identification of true Pareto-optimal solution sets, only approximated solution sets, potentially affecting evaluators' judgments on solution quality and diversity.

---

> ### Comment · Area_Chair_aPW4 · 2023-08-03
> **Posted review for the wrong paper?**
>
> Hi Reviewer EY2k,
>
> It looks like this review is for a different paper? Did you accidentally paste the wrong one in?
>
> Thanks for checking!

---

> ### Author Rebuttal · Authors · 2023-08-10
>
> We appreciate the reviewer for the valuable comments, and finding our method novel, experiment comprehensive, and results advantegous. We hope the point-to-point response below would address the remaining concerns.
>
> **To Weakness 1:** Our method is designed for general MOCOPs. Other advancing techniques can be integrated to address more complex or higher-dimensional MOCOP.
>
> Regarding underfitting and overfitting, we observe from the training curves (see Figures 4(b) and 6) that our model has converged. Additionally, we evaluate our model on 200 randomly generated instances for each problem, as well as on larger-scale and real-world benchmark instances. The results demonstrate the desirable generalization ability of our model.
>
> We would like to supplement the discussion on weight vector selection as follows.
>
> Different weight vectors will lead to different solution distributions. EMNH can utilize proper weight selection methods to obtain a more uniform Pareto front, since EMNH can flexibly tackle arbitrary weight vectors. Specifically, two methods are used for demonstration. (1) A weight assignment method used in PMOCO [1]. (2) We design a scaling weight assignment (SWA) method as follows. Each of the uniform weight vectors $\lambda$ is scaled by $f'$ as $\lambda_m/f'_m$ and normilized into $[0,1]^M$, where $f'_m$ is estimated on a validation dataset associated with $\lambda=(1/M,\dots,1/M)$.
>
> We have supplemented the results on tri-TSP with asymmetric Pareto fronts, as presented in the PDF in "Global Rebuttal". For tri-TSP instances, the coordinates for the three objectives are randomly sampled from $[0,1]^2$, $[0,0.5]^2$, $[0,0.1]^2$, respectively. According to the first method, the non-uniform weight vectors which are obtained by multiplying uniform weight vectors by (1,2,10) element-wise and then normalizing them back to $[0,1]^3$. The results show that two weight vector selection methods can produce a relatively uniform Pareto front.
>
> Furthermore, in our framework, more complex and effective weight vector selection methods [2] can be adapted to handle irregular Pareto fronts.
>
>
> [1] Pareto Set Learning for Neural Multi-Objective Combinatorial Optimization, ICLR, 2022.
>
> [2] A Survey of Weight Vector Adjustment Methods for Decomposition based Multi-objective Evolutionary Algorithms, IEEE TEVC, 2020.
>
>
> **To Weakness 2:** We have tried to provide a comprehensive description of the model architecture in Section 4.1 and Appendix A, along with detailed information on hyperparameter settings and the experimental setup in Section 5.1 and Appendix D. Furthermore, we have made our source code and datasets used in the experiments available, enabling readers to replicate and validate the results.
>
>
> **To Question 1:** We have focused on utilizing POMO and Reptile, as they are currently considered state-of-the-art methods in the field of single-objective neural combinatorial optimization and meta-learning, respectively. To ensure fairness in our comparisons, all the neural heuristics we compared with employed POMO as their single-objective base model, while the MDRL method we compared with also utilized Reptile. In future research, we plan to explore more advanced meta-learning algorithms and more powerful base models to further enhance the performance of our method.
>
>
> **To Question 2:** We have conducted experiments on three classic MOCOPs: MOTSP, MOCVRP, and MOKP. The MOTSP consists of four different types. Each problem includes at least three scales, and even five scales for MOTSP.
>
> In our future work, we plan to expand the scope of our method by testing it on more complex MOCOPs. We also aim to improve its generalization capabilities across various problem types and scales by leveraging leading techniques such as pre-training and knowledge distillation.
>
>
> **To Limitation 1:** Our method may require a large amount of training data and computational resources. However, the offline computational cost is affordable and worthy in practical applications because a well-trained model can quickly generate high-quality solutions for similar problem instances within a specific problem class. This capability makes our method highly valuable in scenarios where efficiency and quality are crucial considerations.
>
>
> **To Limitation 2:** Our method is designed to address a wide range of MOCOPs, including those with nonlinear or non-convex objective functions. The objective function of a combinatorial problem is inherently non-convex. In the case of the Tchebycheff-decomposed subproblem (as described in Appendix J.2), it even becomes nonlinear.
>
> To handle more complex problem features, our method can be combined with other techniques. For instance, when dealing with complex constraints, our method can incorporate a constraint-handling technique [2] to effectively handle them. Additionally, for dynamic problems, our method can leverage a spatio-temporal neural model [3] to enhance its performance in such scenarios.
>
> [2] Deep Reinforcement Learning with Two-Stage Training Strategy for Practical Electric Vehicle Routing Problem with Time Windows, PPSN, 2022.
>
> [3] Solving Dynamic Traveling Salesman Problems With Deep Reinforcement Learning, IEEE TNNLS, 2023
>
> **To Limitation 3:** Due to the NP-hardness of MOCOPs, finding the exact Pareto set within acceptable computational time is often impractical, especially for large-scale problems. Therefore, it is crucial to design approximate methods that can provide near-optimal solutions within a reasonable timeframe.

---

> > ### Comment · Reviewer_EY2k · 2023-08-15
> > **Official Comment by Reviewer EY2k**
> >
> > The reviewer appreciates the authors' detailed response. After reading the rebuttal, I have decided to maintain the previous rating.

---

> > > ### Author Response · Authors · 2023-08-15
> > >
> > > We appeciate the reviewer for acknowledging our work and response.

---

### Official Review · Reviewer_qYcb · 2023-07-06

**Soundness:** 3 good
**Presentation:** 3 good
**Contribution:** 3 good
**Rating:** 6
**Confidence:** 3

**Summary:**

The paper proposes efficient meta neural heuristic (EMNH) for solving multi-objective combinatorial optimization problems (MOCOP). The paper provides novel scaled sampling method for stability and a hierarchical fine-tune method for sub-task specific performance improvement over MDRL. The idea is sound and the experiments yield better performance than the state-of-the-art methods for MOCOP.

**Strengths:**

The idea of using multi-task body with different heads in meta-learning is sound.
The experiments are detailed and convincing.


**Weaknesses:**

I’m not an expert in MOCOP problem, as far as I’m concerned, I detect no weakness of this paper.

**Questions:**

1. According to Table 2, although the traditional heuristics may yield the best performance, it would take them much more time to solve the problem. Given the same amount of fine-tuning time, what is the performance of the proposed MOCOP method EMNH? (i.e. set K to a large number).

**Limitations:**

The proposed EMNH provides no guarantee for the pareto front.

---

> ### Author Rebuttal · Authors · 2023-08-10
>
> We appreciate the reviewer for the valuable comments, and finding our experiments detailed and convincing. Our EMNH method, when provided with a sufficiently large value of $K$ (the number of fine-tuning steps), may not outperform traditional strong solvers like WS-LKH. We conducted a study to examine the impact of $K$ on performance in the original submission, and the results are presented in Figure 4(c) and Figure 7. It is evident that the model has almost converged when $K$ equals 20. Consequently, further increasing $K$ has little effect on performance, which aligns with our additional tests. It is worth noting that the model is fine-tuned on the fine-tuning instances for a given weight vector, rather than the test instances.
>
> Nevertheless, our EMNH approach excels in generating a desirable Pareto set within a short solving time. It can also serve as an initial solution generator, which can be further enhanced using active search techniques [1] or traditional heuristics. However, we understand and acknowledge the concern of the reviewer. To further improve the performance of EMNH, we also plan to investigate more advanced meta-learning algorithms and employ more powerful base models, as outlined in the future work section of our paper.
>
> [1] Efficient Active Search for Combinatorial Optimization Problems, ICLR, 2022.

---

> > ### Comment · Reviewer_qYcb · 2023-08-14
> > **Reply to authors**
> >
> > Dear authors, thanks for your time in rebuttal. I have no more questions and I raise my score to 6.

---

> > > ### Author Response · Authors · 2023-08-14
> > >
> > > We appeciate the reviewer for acknowledging our work and response.

---

### Official Review · Reviewer_d8Lb · 2023-07-06

**Soundness:** 3 good
**Presentation:** 3 good
**Contribution:** 3 good
**Rating:** 8
**Confidence:** 4

**Summary:**

This work proposes EMNH, an efficient meta neural heuristic, for solving multi-objective combinatorial optimization problems. It builds a single meta model to tackle different trade-offs among multiple objectives during training, which can be efficiently fine-tuned into specialized submodels to solve different trade-offed subproblems. The main contributions are: 1) a multi-task meta model with parameter-shared body and task-related heads that can handle different trade-offs at the same time; 2) a scaled symmetric sampling method to stabilize the multi-objective training with imbalanced objectives; and 3) a hierarchical fine-tuning method to gradually fine-tune a set of submodel with much fewer steps. Experimental results show that the proposed EMHN method can outperform other neural combinatorial optimization methods on different combinatorial optimization problems such as MOTSP, MOCVRP, and MOKP.

**Strengths:**

+ This paper is well-written and easy to follow.

+ Multi-objective combinatorial optimization is important for real-world applications. Neural heuristic is a promising approach to tackle this problem, but only a few methods have been proposed recently. This work is a timely contribution to a promising research direction.

+ The proposed method obtains promising performances on various multi-objective combinatorial optimization problems.

**Weaknesses:**

I have the following concerns for the proposed method:

**1. Runetime and Efficiency of Fine-tune**

The runtime of the fine-tuning approach is not clearly discussed in the paper. In my understanding, for solving a new problem instance, the meta-learning based methods need to first fine-tune the meta model into different submodels, and then run each submodel to generate corresponding approximate solutions. In other words, the prediction is not zero-shot inference. However, in all experiments, the reported runtimes for MDRL and EMNH are similar to DRL-MOA and PMOCO which support zero-shot inference. Why the cost of fine-tuning is not included in the runtime?

**2. Structure of Multi-Task Meta Model**

The proposed multi-task meta model has a single model body that is shared by all tasks and specialized heads for different tasks. Recent work has also investigated fine-tuning a small part of the neural combinatorial optimization model to improve the performance of each instance for single-objective problems [1]. What is the advantage of the proposed model structure for multi-objective optimization? Is there any guideline for model design?

[1] Efficient Active Search for Combinatorial Optimization Problems, ICLR 2022.

**3. Solution Distribution**

EMNH mainly uses the weighted-sum strategy to decompose a multi-objective optimization problem into multiple subproblems. According to PMOCO [1], different decomposition methods will lead to very different solution distributions on a given problem instance, especially those with more than two objectives. In addition, as reported in a current work [2], the decomposition-based PMOCO will generate redundant solutions for different preferences. Will EMNH also have these two issues? Can these issues be addressed by the scaled symmetric weight sampling method?

[2] Pareto Set Learning for Neural Multi-Objective Combinatorial Optimization, ICLR 2022.

[3] Graph Learning Assisted Multi-Objective Integer Programming, NeurIPS 2022.

**4. Fine-Tuning PMOCO**

As reported in this work, PMOCO usually has a promising zero-shot prediction performance, but it cannot be further improved with the fine-tuned approach. In my understanding, PMOCO is also a variant of the AM structure in POMO. According to recent works [1], only fine-tuning a small part of the model parameters in POMO can significantly improve the performance of single-objective optimization. It is interesting to know why fine-tuning does not work for PMOCO.

**Questions:**

- Please address the concerns raised in Weaknesses.

**Limitations:**

The limitation of the proposed method has been discussed in the conclusion section.

---

> ### Author Rebuttal · Authors · 2023-08-10
>
> We appreciate the reviewer for the valuable comments, and considering our paper well written with timely contribution and promising performance. We hope the point-to-point response below would address the remaining concerns.
>
> **To Weakness 1: Runtime and Efficiency of Fine-tune.** The fine-tuning time is not included in the runtime, as the runtime refers to the inference time on test instances. It's important to note that all our inference results are zero-shot on test instances. As explained in Section 3.3, a submodel corresponding to a specific weight vector is fine-tuned from the well-trained meta-model using fine-tuning instances, rather than using the test instances as done in [8]. This means that the submodel hasn't seen the test instances before the inference. Therefore, the runtime represents the zero-shot inference time. In this sense, the runtime of EMNH and MDRL is also similar to that of DRL-MOA and PMOCO.
>
> [8] Meta-learning-based Deep Reinforcement Learning for Multiobjective Optimization Problems, IEEE TNNLS, 2022.
>
> **To Weakness 2: Structure of Multi-Task Meta Model.** One advantage of our model over [1] is faster inference due to its simpler architecture. In [1], EAS is instance-specific, whereas our model is task-specific, meaning it can infer a batch of instances from a similar task in parallel. Unlike [1], our meta-model has the same architecture as the single-objective neural model, eliminating the need for additional model designs. The multi-task model only requires multiple heads to operate in parallel and does not need to be saved after training the meta-model.
>
> Our model design follows the guideline of feature reuse in meta-learning [4]. As discussed in [4], only the head of the meta-model is updated to align with a specific task, while the body remains reusable for all tasks. This principle has inspired us to propose a multi-task model that accelerates the training of the meta-model.
>
> [1] Efficient Active Search for Combinatorial Optimization Problems, ICLR, 2022.
>
> [4] Rapid learning or feature reuse? towards understanding the effectiveness of MAML, ICLR, 2020
>
> **To Weakness 3: Solution Distribution.** Similar to other decomposition-based methods like PMOCO, EMNH also faces the challenge of different decomposition methods resulting in different solution distributions. However, this issue can be mitigated by employing appropriate weight assignment methods, as demonstrated in PMOCO. EMNH offers the flexibility to handle arbitrary weight vectors. Specifically, when the approximate scales of different objectives are known, we can normalize them to [0,1] to achieve a more uniform Pareto front. Alternatively, we can adjust the weight assignment to generate a more uniform Pareto front. In future work, more advanced weight assignment methods [5] could be explored to address irregular Pareto fronts beyond the scope of this paper.
>
> Furthermore, like other decomposition-based methods, EMNH may generate redundant solutions for different weight vectors. This is an inherent limitation of the decomposition approach. To promote greater diversity in solutions, one potential direction is to consider a diversity indicator [6] or divide the objective space into regions [7].
>
> The scaled symmetric sampling method is proposed to stabilize the training process and can be directly used to address these issues. We also attempt to apply the scaling operation to the weight assignment during inference to alleviate the first issue. Specifically, each uniform weight vector $\lambda$ is scaled by $f'$ as $\lambda_m/f'_m$ and normalized to $[0,1]^M$. Here,$f'_m$ is estimated using a validation dataset associated with $\lambda=(1/M,\dots,1/M)$. The advantage of this scaling weight assignment (SWA) method is that it does not require prior problem information.
>
> In the "Global Rebuttal" PDF, we have included additional results on tri-TSP instances with asymmetric Pareto fronts. For these instances, the coordinates for the three objectives are randomly sampled from $[0,1]^2$, $[0,0.5]^2$, $[0,0.1]^2$, respectively. The results demonstrate that EMNH-SWA effectively produces a more uniform Pareto front. Compared to a scaling weight method with prior knowledge, where uniform weight vectors are element-wise multiplied by (1,2,10) and then normalized back to $[0,1]^3$, EMNH-SWA achieves desirable performance.
>
> [5] A Survey of Weight Vector Adjustment Methods for Decomposition based Multi-objective Evolutionary Algorithms, IEEE TEVC, 2020.
>
> [6] Performance Indicators in Multiobjective Optimization, EJOR, 2021.
>
> [7] Improving Pareto Local Search Using Cooperative Parallelism Strategies for Multiobjective Combinatorial Optimization, IEEE TCYB, 2022
>
> **To Weakness 4: Fine-Tuning PMOCO.** Fine-tuning is not effective for PMOCO, since PMOCO has already converged for the subproblems corresponding to the given weight vectors. As mentioned earlier, our fine-tuning process is performed on fine-tuning instances, not on test instances. This approach differs from the 'fine-tuning' executed on each test instance, also known as active search [1]. Hence, our fine-tuning does not improve the performance of the well-trained PMOCO model for zero-shot inference on test instances.

---

> > ### Comment · Reviewer_d8Lb · 2023-08-15
> > **Thank you for the detailed response.**
> >
> > Thank you for your detailed response, and all my concerns have been properly addressed. I keep my positive score (6) and lean toward accepting this paper.

---

> > > ### Author Response · Authors · 2023-08-15
> > >
> > > We appreciate the reviewer for acknowledging our work and rebuttal.

---

> > > ### Author Response · Authors · 2023-08-21
> > >
> > > Dear Reviewer d8Lb,
> > >
> > > We again appreciate your time for reviewing our paper. We realize that some additional results may faciliate further recognizing the value of our work comprehensively. Thus, we have conducted a supplementary study about the fune-tuning efficiency.
> > >
> > > **Additional Response to Weakness 1: Trade-off Between Fine-tuning Efficiency and Performance.** For a given weight vector, EMNH fine-tunes the meta-model to derive a submodel to solve the corresponding subproblem. We study another two (relatively) lightweight fine-tuning methods, including only updating the head parameter (denoted as EMNH-FH) according to feature reuse [1] and only updating the decoder parameter (denoted as EMNH-FD) like PMOCO [2]. These two methods even allow us to only fine-tune and store parts of the original submodels, i.e., $N$ heads or $N$ decoders, thereby being more computationally efficient. Meanwhile, such benefit may bring about performance sacrifices in some cases. We report the results and the parameter numbers of various parts of the model in the table below. The lightweight fine-tuning has slightly inferior performance compared with the original EMNH in most cases except on Bi-CVRP ($n$=100). Generally,  the more lightweight of the fine-tuning, the more performance deterioration (i.e.,EMNH-FH v.s. EMNH-FD as displayed in the table below, where FH is more light than FD). However, these lightweight fine-tuning methods can be used as alternatives when the computational and memory resources are limited.
> > >
> > > Moreover, same as EMNH, EMNH-FH can also generate much more dense Pareto solutions to improve the performance via increasing weight vectors and corresponding fine-tuned heads. We have plotted the generated Pareto fronts with 105, 300 and 1035 weight vectors on Tri-TSP-1 which verified the above point. However,  we will supplement this result in the final version since the "Global Rebuttal" PDF containing figures is not allowed to updated or uploaded at this moment.
> > >
> > > **Table: Results of lightweight fine-tuning methods.**
> > >
> > > Bi-CVRP ($n$=20)|HV|Gap
> > > ---|:---:|:---:
> > > PMOCO-Aug|0.4294|0.19%
> > > MDRL-Aug|0.4292|0.23%
> > > EMNH-Aug|0.4302|0.00%
> > > EMNH-FD-Aug|0.4299|0.07%
> > > EMNH-FH-Aug|0.4298|0.09%
> > >
> > > Bi-CVRP ($n$=100)|HV|Gap
> > > ---|:---:|:---:
> > > PMOCO-Aug|0.3966|2.77%
> > > MDRL-Aug|0.4072|0.17%
> > > EMNH-Aug|0.4079|0.00%
> > > EMNH-FD-Aug|0.4082|-0.07%
> > > EMNH-FH-Aug|0.4082|-0.07%
> > >
> > > Tri-TSP-1 ($n$=20)|HV|Gap
> > > ---|:---:|:---:
> > > PMOCO-Aug|0.4712|0.00%
> > > MDRL-Aug|0.4712|0.00%
> > > EMNH-Aug|0.4712|0.00%
> > > EMNH-FD-Aug|0.4710|0.04%
> > > EMNH-FH-Aug|0.4707|0.11%
> > >
> > > Tri-TSP-1 ($n$=100)|HV|Gap
> > > ---|:---:|:---:
> > > PMOCO-Aug|0.4956|0.34%
> > > MDRL-Aug|0.4958|0.30%
> > > EMNH-Aug|0.4973|0.00%
> > > EMNH-FD-Aug|0.4925|0.97%
> > > EMNH-FH-Aug|0.4906|1.35%
> > >
> > > **Table: Parameter numbers of various parts of the model.**
> > >
> > > Bi-CVRP Model|#(Parameters)|
> > > ---|:---:
> > > Whole Model|1287K
> > > Decoder|98K
> > > Head|16K
> > >
> > > Tri-TSP-1 Model|#(Parameters)|
> > > ---|:---:
> > > Whole Model|1303K
> > > Decoder|115K
> > > Head|16K
> > >
> > > **Reference**
> > >
> > > [1] Rapid learning or feature reuse? towards understanding the effectiveness of MAML, ICLR, 2020.
> > >
> > > [2] Pareto Set Learning for Neural Multi-Objective Combinatorial Optimization, ICLR, 2022.

---

> > > > ### Comment · Reviewer_d8Lb · 2023-08-21
> > > > **Thank you for the response.**
> > > >
> > > > Thank you for the response. I appreciate the author's effort in providing further discussion with additional experiments to address my concerns. All my concerns are now fully addressed.
> > > >
> > > > I have also read other reviewers' comments and the author's response in detail. I increase my rating to 8 to fully support accepting this paper.

---

> > > > > ### Author Response · Authors · 2023-08-21
> > > > >
> > > > > We appreciate the reviewer for acknowledging our further response and support our work.

---

### Official Review · Reviewer_R5xR · 2023-07-06

**Soundness:** 3 good
**Presentation:** 3 good
**Contribution:** 3 good
**Rating:** 6
**Confidence:** 2

**Summary:**

The paper introduces a meta neural heuristic in which a meta model is first trained and then fine-tuned with a few steps
to solve corresponding single-objective subproblems. For the training process, a partial architecture-shared multi-task model is leveraged to achieve parallel learning for the meta model, so as to speed up the training. Meanwhile, a scaled symmetric sampling method with respect to the weight vectors is designed to stabilize the training. For the fine-tuning process, an efficient hierarchical method is proposed to systematically tackle all the subproblems.

The article contains a review of related works and preliminaries, then it presents the introduced methodology.
This is followed by the description of experiments, their settings, results, and analysis. The experiments were carried out for
the multi-objective traveling salesman problem, multi-objective capacitated vehicle routing problem, and multi-objective knapsack problem. They showed that the introduced method is able to outperform the state-of-the-art neural heuristics in terms of solution quality and learning efficiency and yield competitive solutions to the strong traditional heuristics while consuming a much shorter time.

The main text is followed by supplementary materials.

**Strengths:**

The introduced methodology is quite advanced and seems to be innovative and successful.
I also like that it was tested on several problems from different domains and besides reporting efficiency and the quality of the found solutions, the required time is reported too. The paper is well written and it is good that it is followed by the supplementary materials. I like that there is a pseudocode in the supplementary materials and the authors declared that the codes for all the methods will be made available.

**Weaknesses:**

The main weakness I found is that the code and the datasets used in experiments are not available, so it is difficult to verify and reproduce the results. However, the authors provided some pseudocode and declared that the code will be made publicly available. Besides, didn't find serious weaknesses of this paper, but it is possible that I didn't understand some parts.

**Questions:**

I don't have any specific questions.

**Limitations:**

The main limitation, according to the authors, is that the method can not guarantee to obtain the exact Pareto front (similar to other neural heuristics). Also, the code and the datasets used in experiments are not available, so it is difficult to verify and reproduce the results. However, the authors provided some pseudocode and declared that the code will be made publicly available. I didn't find other limitations, but it is possible that I didn't understand some parts.

---

> ### Author Rebuttal · Authors · 2023-08-10
>
> We appreciate the reviewer for the valuable comments, and considering our method advanced and our paper well written. Regarding the source code, on the one hand, we have stated clearly in our original submission - ' Our codes for all the methods will be made available'. On the other hand, we will upload our source code and datasets, and share the url with the AC (according to the NeurIPS rebuttal policy). We thank the reviewer for the support again.

---

> > ### Comment · Reviewer_R5xR · 2023-08-10
> >
> > Thank you for the information. I've read the rebuttal and don't have more questions now. I keep my previous decision.

---

> > > ### Author Response · Authors · 2023-08-10
> > >
> > > We appreciate the reviewer for acknowledging our work and rebuttal.

---

### Official Review · Reviewer_TTgk · 2023-07-07

**Soundness:** 3 good
**Presentation:** 2 fair
**Contribution:** 2 fair
**Rating:** 4
**Confidence:** 2

**Summary:**

In order to achieve higher learning efficiency and better solution quality, this paper proposed an efficient meta neural heuristic (EMNH), in which a meta model is first trained and then fine-tuned with a few steps to solve corresponding single-objective subproblems. For the training process, an architecture-shared multi-task model is leveraged to achieve parallel learning for the meta model, so as to accelerate the training; meanwhile, a scaled symmetric sampling method with respect to the weight vectors is designed to stabilize the training. For the fine-tuning process, an efficient hierarchical method is proposed to systematically tackle all the subproblems. The general idea of this paper is clear and logical.

**Strengths:**

1. The methodology part of the paper is clearly described and has a certain degree of logic.

**Weaknesses:**

1. In the Methodology part, this paper proposed three methods for accelerate training, stabilize training and hierarchical Fine-tuning. However, in the Introduction part, the connection between motivation of this paper and the proposed stabilized training method should be clearer described.
2. In the Experimental Results part, ablation experiment should be added in order to verify the effect of the proposed method. To be more specific, as for the experiments towards learning efficiency, the comparison between original method and method without adding stabilized training part should be done.


**Questions:**

1. In the Experimental Results part, the experiment results in tables should be further analyzed.
2. In the Training Efficiency part, more current comparison method should be added in order to support the experimental conclusion with more sufficient and comprehensive experimental results.
3. Are there any control parameters in the proposed method? how sensitive are them?
4. The picture in Figure 4 is too small for clearly understanding of readers.
5. The writing of the paper could be improved for better description and clarification.

**Limitations:**

1. The experimental problems of this method are limited to real-world problems, and further experimental verification should be carried out on synthetic problems.

---

> ### Author Rebuttal · Authors · 2023-08-10
>
> We appreciate the reviewer for the valuable comments, and considering our idea clear and logical. We hope the point-to-point response below would address the remaining concerns.
>
> **To Weakness 1:**  We acknowledge and appreciate the reviewer's concern. It is indeed crucial to establish a clear connection between the motivation of our paper and the proposed stabilized training method. As a result, we will carefully revise and add the following statements to ensure consistency in conveying these points throughout the paper.
>
> '...The recent Meta-DRL (MDRL) [13] has demonstrated the capability to enhance solution quality compared with PMOCO. However, it still faces challenges related to inefficient and unstable training procedures, as well as undesirable fine-tuning processes.'
>
> 'During the meta-learning process, the deviation of a few randomly sampled weight vectors may cause fluctuations to the parameter update of the meta model, leading to the unstable training performance. This motivates us to introduce a stabilized training method.'
>
>
> **To Weakness 2:** Thank you for your suggestion regarding the ablation study. In our original submission, we have already conducted an ablation experiment specifically focusing on the proposed scaled symmetric sampling method. The detailed results can be found in Section 5.3 and Appendix F.2.
>
> To be specific, we compared EMNH with two variants EMNH-R (EMNH with random sampling) and EMNH-S (EMNH with symmetric sampling). EMNH-R reflects the overall effect of the complete scaled symmetric sampling method, while EMNH-S isolates the effect of the corresponding scaling operation. The results clearly demonstrate that the proposed scaled symmetric sampling method effectively stabilizes the training process.
>
>
> **To Question 1:**  Thanks for raising the concern on the experimental results anaysis. We acknowledge that the analysis provided in the original submission was concise due to rich presented results and limited space. According to the suggestions, we will add more detailed analysis in the final version. For example, the analysis about the results in Figure 4(c) will be supplemented as follows.
>
> '...Notably, we observed that EMNH with a few fine-tuning steps (e.g., $K = 5$) generally outperforms PMOCO in most cases, as demonstrated in Appendix F. It is important to note that our fine-tuning process is performed individually for each weight vector on dedicated fine-tuning instances. As a result, the fine-tuning does not enhance the performance of the well-trained PMOCO model for zero-shot inference on test instances. This finding indicates that PMOCO has already converged for the subproblems corresponding to the given weight vectors...'
>
> **To Question 2:**  The objective of this paper is to evaluate neural solvers or heuristics for MOCOP. We have carefully selected MDRL [1], PMOCO [2], and DRL-MOA [3] as the neural solvers for comparison. The selection of these representatives is based on the following reasons. As show in the Related Works section, DRL-MOA is acknowledged as a well-known classic neural heuristic for MOCOP. MDRL and PMOCO are recognized as state-of-the-art neural heuristics, surpassing previous approaches, including DRL-MOA.
>
> From our perspective, the experimental results provide compelling evidence to support our conclusion. The original submission includes a comprehensive analysis of training efficiency in Section 5.3 and Appendix F.1. The results clearly demonstrate that EMNH achieves a training time of approximately 1/$\tilde{N}$ compared to MDRL. Moreover, the total training time of EMNH is comparable to that of PMOCO. In contrast, DRL-MOA requires training multiple models, resulting in significantly more time, which is less competitive.
>
> **To Question 3:** Thank you for raising the concern about the parameters. We have addressed this issue in Section 5.1, where all the control hyperparameters are listed. Most of these parameters are adopted from previous works, including the learning rate [1,2], the meta-learning rate $\epsilon$ [1], the number of gradient steps for the inner-loop update $T_u$ [1], the batch size $B$ [2], and the number of weight vectors $N$ [2].
>
> We specifically investigated those parameters used in our EMNH, which include the number of sampled weight vectors $\tilde{N}$, the scalarization method, and the number of fine-tuning steps at each level $K$. The sensitivity analysis of $\tilde{N}$ and the scalarization method can be found in Appendix J, while the analysis of $K$ can be found in Appendix F.3. In summary, our findings suggest that setting $\tilde{N}=M$ and using the weighted-sum scalarization method yield desirable results. Additionally, we observed that the model has nearly converged when $K=20$.
>
> **To Question 4:**  Thanks for pointing it out. We appreciate the suggestion, and will consider enlarging the picture in Figure 4 if we have sufficient space in the final version.
>
> **To Question 5:** Thanks for the suggestion. We will thoroughly proofread and revise the writing in the final version.
>
> **To Limitation 1:** Thanks for raising this concern. We want to clarify that our experiments were conducted on a diverse set of problem instances, including both real-world problems and synthetic instances. For the real-world problems, we utilized the KroAB instances from TSPLIB, as indicated in Table 8. These instances have also been used in previous works [1][3].
>
> In addition to the real-world problems, we also incorporated randomly generated instances for our experiments. The details and results of these instances can be found in Tables 1 and 2, which are consistent with the approaches taken in prior studies [1][2][3].
>
> [1] Meta-learning-based Deep Reinforcement Learning for Multiobjective Optimization Problems, IEEE TNNLS, 2022.
>
> [2] Pareto Set Learning for Neural Multi-Objective Combinatorial Optimization, ICLR, 2022.
>
> [3] Deep Reinforcement Learning for Multiobjective Optimization, IEEE TCYB, 2021.

---

> > ### Comment · Reviewer_TTgk · 2023-08-15
> >
> > I appreciate the authors' comprehensive response, which effectively addressed the majority of my inquiries. Nevertheless, given the modest degree of enhancement and the relatively minor contributions, I am inclined to uphold my initial assessment.

---

> > > ### Author Response · Authors · 2023-08-16
> > >
> > > We appreciate the reviewer for the feedback. And we hope that the response below would address the outstanding concerns. In this regard, we would like to re-highlight our contribution and enhancement as follows.
> > >
> > > 1. **Contribution.** It is a known fact that there has been a growing trend to exploit neural heuristic based on deep reinforcement learning to solve MOCOP. However, the prior works of this line, including the state-of-the-art MDRL and PMOCO, still struggle to achieve high learning efficiency and solution quality. We thereby propose an efficient meta neural heuristic (EMNH) to push the boundary of this line of research. EMNH outperforms state-of-the-art neural heuristics in terms of learning efficiency and solution quality (see Figure 4, Table 1, and Table 2). Meanwhile, EMNH can produce competitive solutions to the strong traditional heuristics with much shorter solving time, e.g., EMNH's Gap 0.00% with 1.7 minutes v.s. WS-LKH's Gap -0.11% with 1.8 hours for Bi-TSP-1 ($n$=50) in Table 1.
> > > 2. **Novelty.** We propose an accelerated training method via feature reuse and architecture-shared multi-task learning, a stabilized training method via scaled symmetric sampling, and an efficient hierarchical fine-tuning method.
> > > 3. **Enhancement.** In terms of **learning efficiency**, our EMNH only spends about $1/\tilde{N}$ training time ($\tilde{N}$ is set to 6 at most in Figure 4(a)) of the state-of-the-art MDRL; our EMNH achieves the stablest and best training performance compared with MDRL and other baselines, as shown in Figure 4(b); and our EMNH attains higher performance than MDRL with approximately equal total fine-tuning steps, e.g., EMNH's HV 0.6585 v.s. MDRL's HV 0.6441 for $K=1$ in Figure 4(c). In terms of **solution quality**, EMNH outperforms other neural heuristics, especially demonstrating a significant advantage over the state-of-the-art PMOCO, e.g., EMNH's Gap 0.17% v.s. PMOCO's Gap 4.19% for Bi-CVRP ($n$=100) in Table 2.
> > >
> > > In summary, we believe that our contribution and enhancement are significant in the field of neural MOCO, which could also inspire the subsequent works. Notably, as acknowledged by Reviewer d8Lb with high confidence, 'This work is a timely contribution to a promising research direction.'
> > >
> > > Please feel free to let us know if the reviewer still has any other concrete or specific concerns. We are happy to take them as suggestions to further improve our work.

---

> > > > ### Author Response · Authors · 2023-08-21
> > > >
> > > > Dear Reviewer TTgk,
> > > >
> > > > We appreciate your time for reviewing our paper. The comments are indeed helpful for improving our work. Please kindly let us know if you still have any other concerns, so that we could respond to them timely, given that the reviewer-author discussion deadline is approaching soon. Thanks for your understanding.
> > > >
> > > > Authors of 12293

---

> > > ### Author Response · Authors · 2023-08-21
> > >
> > > Dear Reviewer TTgk,
> > >
> > > We again appreciate your time for reviewing our paper. We conducted an additional study about the lightweight fune-tuning which may raise the **contribution** and **enhancement** of our work as shown below.
> > >
> > > **Trade-off Between Lightweight Fine-tuning and Performance.** For a given weight vector, EMNH fine-tunes the meta-model to derive a submodel to solve the corresponding subproblem. We study another two (relatively) lightweight fine-tuning methods, including only updating the head parameter (denoted as EMNH-FH) according to feature reuse [1] and only updating the decoder parameter (denoted as EMNH-FD) like PMOCO [2]. These two methods even allow us to only fine-tune and store parts of the original submodels, i.e., $N$ heads or $N$ decoders, thereby being more computationally efficient. Meanwhile, such benefit may bring about performance sacrifices in some cases. We report the results and the parameter numbers of various parts of the model in the table below. The lightweight fine-tuning has slightly inferior performance compared with the original EMNH in most cases except on Bi-CVRP ($n$=100). Generally,  the more lightweight of the fine-tuning, the more performance deterioration (i.e.,EMNH-FH v.s. EMNH-FD as displayed in the table below, where FH is more light than FD). However, these lightweight fine-tuning methods can be used as alternatives when the computational and memory resources are limited.
> > >
> > > Moreover, same as EMNH, EMNH-FH can also generate much more dense Pareto solutions to improve the performance via increasing weight vectors and corresponding fine-tuned heads. We have plotted the generated Pareto fronts with 105, 300 and 1035 weight vectors on Tri-TSP-1 which verified the above point. However,  we will supplement this result in the final version since the "Global Rebuttal" PDF containing figures is not allowed to updated or uploaded at this moment.
> > >
> > > **Table: Results of lightweight fine-tuning methods.**
> > >
> > > Bi-CVRP ($n$=20)|HV|Gap
> > > ---|:---:|:---:
> > > PMOCO-Aug|0.4294|0.19%
> > > MDRL-Aug|0.4292|0.23%
> > > EMNH-Aug|0.4302|0.00%
> > > EMNH-FD-Aug|0.4299|0.07%
> > > EMNH-FH-Aug|0.4298|0.09%
> > >
> > > Bi-CVRP ($n$=100)|HV|Gap
> > > ---|:---:|:---:
> > > PMOCO-Aug|0.3966|2.77%
> > > MDRL-Aug|0.4072|0.17%
> > > EMNH-Aug|0.4079|0.00%
> > > EMNH-FD-Aug|0.4082|-0.07%
> > > EMNH-FH-Aug|0.4082|-0.07%
> > >
> > > Tri-TSP-1 ($n$=20)|HV|Gap
> > > ---|:---:|:---:
> > > PMOCO-Aug|0.4712|0.00%
> > > MDRL-Aug|0.4712|0.00%
> > > EMNH-Aug|0.4712|0.00%
> > > EMNH-FD-Aug|0.4710|0.04%
> > > EMNH-FH-Aug|0.4707|0.11%
> > >
> > > Tri-TSP-1 ($n$=100)|HV|Gap
> > > ---|:---:|:---:
> > > PMOCO-Aug|0.4956|0.34%
> > > MDRL-Aug|0.4958|0.30%
> > > EMNH-Aug|0.4973|0.00%
> > > EMNH-FD-Aug|0.4925|0.97%
> > > EMNH-FH-Aug|0.4906|1.35%
> > >
> > > **Table: Parameter numbers of various parts of the model.**
> > >
> > > Bi-CVRP Model|#(Parameters)|
> > > ---|:---:
> > > Whole Model|1287K
> > > Decoder|98K
> > > Head|16K
> > >
> > > Tri-TSP-1 Model|#(Parameters)|
> > > ---|:---:
> > > Whole Model|1303K
> > > Decoder|115K
> > > Head|16K
> > >
> > > **Reference**
> > >
> > > [1] Rapid learning or feature reuse? towards understanding the effectiveness of MAML, ICLR, 2020.
> > >
> > > [2] Pareto Set Learning for Neural Multi-Objective Combinatorial Optimization, ICLR, 2022.

---

### Author Rebuttal · Authors · 2023-08-10

Many thanks for all reviewers' constructive and valuable comments. Following their suggestions, we have made the following main revisions:

1. **Motivation:** We have revised some descriptions to make the connection between our motivation and the proposed method clearer according to the comments of Reviewer TTgk.

2. **Method:** We have supplemented more details of our method, including the model design, fine-tuing process, and inference process.

3. **Experiment:** We have conducted the experiments about the solution distribution according to the suggestions of Reviewer d8Lb.

4. **Addition:** We have further clarified some results of the ablation and hyperparameter study.

---

### Decision · Program_Chairs · 2023-09-21

**Decision:**

Accept (poster)

**Comment:**

All of the reviewers except TTgk agree that the paper should be accepted and reviewer TTgk did not have major concerns - their remaining concerns were around the "modest degree of enhancement and the relatively minor contributions". Other reviewer's found this "This work is a timely contribution to a promising research direction." Given the lack of major concerns and general enthusiasm for the paper by the reviewers, I'm inclined to accept the paper.